# Test-time Offline Reinforcement Learning on Goal-related Experience

**Marco Bagatella** [* 1 2]  **Mert Albaba** [* 1 2]  **Jonas Hübotter** [1]  **Georg Martius** [2 3]  **Andreas Krause** [1]

## Abstract

Foundation models compress a large amount of information in a single, large neural network, which can then be queried for individual tasks. There are strong parallels between this widespread framework and offline goal-conditioned reinforcement learning algorithms: a universal value function is trained on a large number of goals, and the policy is evaluated on a single goal in each test episode. Extensive research in foundation models has shown that performance can be substantially improved through test-time training, specializing the model to the current goal. We find similarly that test-time offline reinforcement learning on experience related to the test goal can lead to substantially better policies at modest compute costs. We propose a novel self-supervised data selection criterion, which selects transitions from an offline dataset according to their relevance to the current state and quality with respect to the evaluation goal. We demonstrate across a wide range of high-dimensional loco-navigation and manipulation tasks that fine-tuning a policy on the selected data for a few gradient steps leads to significant performance gains over standard offline pre-training. Our *goal-conditioned test-time training* (**GC-TTT**) algorithm applies this routine in a receding-horizon fashion during evaluation, adapting the policy to the current trajectory as it is being rolled out. Finally, we study compute allocation at inference, demonstrating that, at comparable costs, GC-TTT induces performance gains that are not achievable by scaling model size.

---

[*]Equal contribution  [1] ETH Zurich, Zurich, Switzerland [2]Max Planck Institute for Intelligent Systems, Tubingen, Germany [3]University of Tubingen, Tubingen, Germany. Correspondence to: Marco Bagatella <mbagatella@ethz.ch>.

*Proceedings of the $43^{rd}$ International Conference on Machine Learning*, Seoul, South Korea. PMLR 306, 2026. Copyright 2026 by the author(s).

## 1 Introduction

Machine learning models are largely static: after a computationally expensive training phase, inference traditionally involves a single forward pass (or multiple, in the case of autoregressive models), without any further parameter updates. This framework is widely adopted across modalities and domains, from early works on image classification (LeCun et al., 1998; He et al., 2016) to many modern vision/language models (Brown et al., 2020; Rombach et al., 2022). However, perfectly imitating training data with a neural network is challenging, and predictions of neural networks are often noisy and imprecise. Consequently, base models are often specialized to down-stream tasks through fine-tuning (Hu et al., 2022; Kim et al., 2022; Black et al., 2024). More recently, across (self-)supervised vision and language tasks, several works improve performance by specializing the model to an individual task, either through in-context learning (Brown et al., 2020) or test-time training (e.g., Sun et al., 2020; Hardt & Sun, 2024; Hübotter et al., 2025). In contrast, in offline reinforcement learning, we face the additional challenge that directly imitating previous experience is generally not optimal for achieving the current goal, either because previous experience was suboptimal or because it was aiming to achieve a different goal. While the dynamic conditioning of a learned policy at test-time has been explored in hierarchical methods (Nachum et al., 2018; Eysenbach et al., 2019; Park et al., 2023), the weights of the policy itself remain generally frozen during evaluation.

Whereas existing offline RL methods freeze policy parameters once training ends, we study the test-time training of goal-conditioned policies. The standard pipeline of offline goal-conditioned reinforcement learning involves (1) a (pre-)training phase, in which a policy learns to reach *arbitrary* goals, often through relabeling or self-supervision, and (2) an inference phase, in which the policy is queried to achieve *one* specific goal. We show that **specializing the policy to an individual goal at test-time significantly improves its performance**, without leveraging any information beyond the pre-training dataset and the pre-trained agent.

We propose *Goal-Conditioned Test-Time Training* (**GC-TTT**), which fine-tunes the base policy at test-time on goal-related experience from the pre-training dataset.[1]

---

[1]While we propose using the pre-training dataset, leveraging privileged or auxiliary data is also possible.

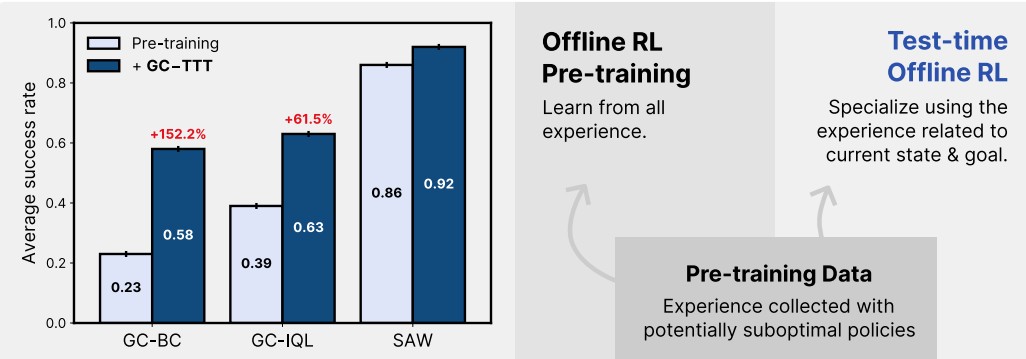

*Figure 1.* We introduce test-time training in the context of offline goal-conditioned reinforcement learning. The same data used for pre-training is filtered and leveraged to improve the policy locally during evaluation. This results in significant performance gains in standard benchmarks (left) when combined with common offline RL backbones, GC-BC, GC-IQL, and SAW.

GC-TTT selects experience according to a natural notion of relevance and optimality, ensuring that it is (1) related to the agent's current state, and (2) optimal with respect to a bootstrapped value function estimate (i.e., a *critic*). Based on this goal-related experience, GC-TTT efficiently updates the actor through few gradient steps according to standard policy learning objectives. We repeat this process in a receding-horizon fashion to periodically and dynamically adapt the policy to the current trajectory.

We demonstrate how GC-TTT improves performance in standard offline goal-conditioned benchmarks, suggesting that existing methods that learn to achieve arbitrary goals fail to retrieve the optimal multi-goal policy, but can be efficiently adapted to retrieve a good single-goal policy. We show that GC-TTT can learn from both expert and play-like data, and additionally derive a variant, which does not require a learned critic and retains good performance on expert data. Both variants are agnostic to the backbone RL algorithm. Within these settings, we ablate the frequency of test-time training and further investigate the compute allocation at test-time, comparing the cost of test-time training against increased model sizes.

We thus make the following contributions:

- We propose a test-time training framework for goal-conditioned policies.

- We develop GC-TTT, a practical algorithm for dynamically training on goal-related experience during evaluation.

- We demonstrate significant performance gains on standard benchmarks when applying goal-conditioned test-time training on top of existing algorithms.

- We demonstrate that GC-TTT significantly outperforms existing algorithms even when inference FLOPs are matched by scaling the network sizes of baselines.

## 2 Related Work

**Goal-conditioned reinforcement learning** Reinforcement learning (RL) research primarily builds upon the framework of Markov decision processes (MDPs), which define their objective based on a scalar function of states and action, referred to as a reward function (Sutton & Barto, 2018). While reward functions may be very expressive (Silver et al., 2021), a conditional reward is more flexible and can model a *family* of behaviors. One such approach is goal-conditioned reinforcement learning (GCRL). Here, the agent's objective is to achieve some specified goal which is modeled by a sparse reward, indicating whether the goal is achieved (Andrychowicz et al., 2017; Eysenbach et al., 2022; Ma et al., 2022; Agarwal et al., 2023). The GCRL framework has been remarkably successful when coupled with neural function approximation (Schaul et al., 2015), which is capable of amortizing the enlarged input space of the policy, compared to individual-task RL. As the reward function is often known, several methods for relabeling (Andrychowicz et al., 2017) and self-supervision (Tian et al., 2021) have been proposed to allow off-policy learning for all possible goals from arbitrary experience. Due to the particular structure of the reward function, goal-conditioned RL allows for specific algorithms beyond TD-learning, including contrastive (Eysenbach et al., 2022; Zheng et al., 2024) and quasimetric (Wang et al., 2023) formulations. Furthermore, goal-conditioned algorithms can be easily adapted to the offline setting considered in our work (Ma et al., 2022; Park et al., 2023; 2025).

In both offline and online settings, the goal-conditioned policy is evaluated by commanding a target goal or a subgoal selected by a high-level component (Nachum et al., 2018; Park et al., 2023). The policy parameters then remain unchanged throughout evaluation. Our work investigates efficient training of the policy weights at test-time, and can be combined with any of the abovementioned value-based algorithms.

**Test-time training** In machine learning, models are traditionally trained on a fixed training set and then kept frozen during evaluation. While this has been the standard practice in machine learning for decades, early work has also discussed specializing the model at test-time to each prediction task. First examples of this so-called transductive approach are local learning (Cleveland, 1979; Cleveland & Devlin, 1988; Atkeson et al., 1997) and local fine-tuning (Bottou & Vapnik, 1992). More recently, the idea of test-time training (TTT) (Sun et al., 2020) has regained attention in the context of fine-tuning large foundation models during evaluation (e.g., Krause et al., 2018; 2019; Hardt & Sun, 2024; Sun et al., 2024). TTT on (self-)supervised signals for few gradient steps has since shown success in domains such as control (Hansen et al., 2021), language modeling (Hardt & Sun, 2024; Hübotter et al., 2025; Sun et al., 2024; Bertolissi et al., 2025), abstract reasoning (Akyürek et al., 2025), and video generation (Dalal et al., 2025).

Many standard TTT methods train on carefully selected data from the pre-training dataset (i.e., do not add any new priviledged information; Hardt & Sun, 2024; Hübotter et al., 2025), and several works studied how to optimally select data for imitation (e.g., MacKay, 1992; Hübotter et al., 2024; Bagatella et al., 2025). Test-time-training is tightly connected to meta-learning (Hochreiter et al., 2001), which aims at producing policies that will rapidly adapt to new environments. While GC-TTT does not assume an explicit meta-learning phase, and may be applied to diverse pre-training algorithms, a meta-learning approach for pre-training may be adopted to amortize test-time costs (Finn et al., 2017; Nichol et al., 2018).

**Test-time reinforcement learning** In this work, we study test-time offline RL (TTORL), where the offline dataset contains trajectories from different policies conditioned on different goals. Therefore, unlike in previous work on TTT, this data should not be imitated directly. Despite this challenge, we show that GC-TTT can substantially improve the performance of standard offline RL algorithms. TTRL has been explored when a model of the environment is available, for instance in the context of games (Fickinger et al., 2021) and autonomous driving (Peng et al., 2024). In this case, model rollouts can be leveraged for policy improvement. Our work is closely related to these approaches, but reuses the offline pre-training dataset for test-time policy improvement. On one hand, this requires careful data selection, which can be self-supervised in the goal-conditioned setting we consider, and on the other it removes the need for the environment's dynamics to be known.

A recent work (Opryshko et al., 2025) also leverages test-time compute to improve goal-reaching performance, but it performs subgoal search instead of optimizing policy parameters. As a consequence, since the goal space is smaller, the search procedure of the latter may likely be more efficient and structured; however, the policy's performance will remain limited to that of the best pre-trained goal-conditioned policy. We refer to Appendix E for a broader discussion. Our work is also closely related to concurrent work, which studies a form of test-time online RL (abbreviated TTRL) with language models (Zuo et al., 2025). Unlike their work, we propose to dynamically train during evaluation of a single goal, which we identify as crucial for achieving maximum performance. Intuitively, our work on TTRL combines the pre-training paradigm commonly pursued in GCRL and the standard RL paradigm of continuously training on experience collected for a single task. In GC-TTT, the pre-trained model is specialized to each individual task during evaluation.

## 3 Background

We model the dynamical system as a reward-free Markov decision process $\mathcal{M} = (\mathcal{S}, \mathcal{A}, P, \gamma, \mu_0)$ (Eysenbach et al., 2022), where $\mathcal{S}$ and $\mathcal{A}$ are potentially continuous state and action spaces, $P : \mathcal{S} \times \mathcal{A} \to \Delta(\mathcal{S})$ is a stochastic transition function, $\gamma$ is a discount factor and $\mu_0 \in \Delta(\mathcal{S})$ is an initial state distribution. We introduce a goal space $\mathcal{G}$ and identify it with the state space $\mathcal{G} = \mathcal{S}$ for simplicity, although goal abstraction remains possible. As standard in goal-conditioned settings, we assume the existence of a distance function $d : \mathcal{S} \times \mathcal{G} \to \mathbb{R}$ to determine *goal achievement*, and define a conditional reward function as

$$R(s, g) = \begin{cases} -1 & \text{if } d(s, g) \geq \epsilon \\ 0 & \text{otherwise,} \end{cases} \quad (1)$$

for some small fixed threshold $\epsilon$. In turn, the reward function induces a conditional value function for each policy $\pi : \mathcal{S} \times \mathcal{G} \to \Delta(\mathcal{A})$:

$$V^\pi(s_0 \mid g) = \mathop{\mathbb{E}}_{P,\pi} \left[ \sum_{t=0}^\infty \gamma^t R(s_t, g) \right]$$

$$\text{where} \quad s_{t+1} \sim P(s_t, a_t),\ a_t \sim \pi(s_t \mid g). \quad (2)$$

Intuitively, the value function computes the negative, expected, discounted number of steps required to reach the goal under a given policy. The optimal policy for some goal distribution $\mu_\mathcal{G}$ can then be defined as $\pi^\star = \arg\max_\pi \mathbb{E}_{g \sim \mu_\mathcal{G}, s_0 \sim \mu_0} V^\pi(s_0; g)$, and induces a quasi-metric structure in its value function (Wang et al., 2023). Most practical algorithms optimize over a broad and dense goal distribution $\mu_\mathcal{G}$ (see, e.g., Andrychowicz et al., 2017), but are only deployed to achieve one specific goal during each episode at inference.

**Offline policy (pre-)training** The standard offline goal-conditioned reinforcement learning pipeline pre-trains a policy $\pi$ on an offline dataset $\mathcal{D}$ of trajectories

$(s_0, a_0, s_1, a_1, \dots)$. Most practical methods parameterize the policy as a neural network $\pi_\theta$, and use stochastic optimization to find

$$\theta_{\text{pre}}^\star = \underset{\theta}{\arg\max} \, J_{\text{pre}}(\theta), \tag{3}$$

for a given pre-training objective $J_{\text{pre}}$ (e.g., stochastic value gradients (Heess et al., 2015) or behavior cloning (Ross et al., 2011)). This objective is normally specified as an expectation over the state-goal distribution from the pre-training dataset:

$$J_{\text{pre}}(\theta) = -\mathbb{E}_{s \sim p_s(\cdot|\mathcal{D}), \, g \sim p_g(\cdot|s,\mathcal{D})} \, \mathcal{L}(s, g, \theta), \tag{4}$$

where $p_s$ and $p_g$ are state and goal distributions, respectively. Normally, the loss function $\mathcal{L}$ will also depend on actions sampled from $\mathcal{D}$; however, these actions are naturally those paired with selected states (e.g., when $\mathcal{L}$ is a behavior cloning loss). Except for prioritized sampling schemes (Horgan et al., 2018), $p_s$ is generally uniform; $p_g$ is instead conditioned on $s$, and may sample future goals from the same trajectories, or random ones (Ghosh et al., 2023). $\mathcal{L}$ represents an arbitrary loss function, and commonly lies on a spectrum between supervised learning (behavior cloning) and fully off-policy reinforcement learning. At its core, the objective in Equation 4 aims to find a policy that is optimal *on average* (w.r.t. the goal distribution $p_g$), which may lead to a locally suboptimal solution for *specific goals*, especially in noisy settings or with limited model capacity.

After this training phase, the policy is evaluated on a *single goal per episode*. We study the problem of fine-tuning the pre-trained model during test-time using offline RL to specialize the policy *locally*. We call this setting *test-time offline reinforcement learning* (TTORL). Our method, GC-TTT, specializes the policy to the agent's current state and goal at test-time.

## 4 GC-TTT: Goal-conditioned Test-time Training

We propose to fine-tune the policy dynamically during evaluation, leveraging data from the pre-training dataset $\mathcal{D}$, which is "close" to the agent's current state $s \in \mathcal{S}$ and "optimal" for reaching the agent's current goal $g^\star \in \mathcal{G}$. We denote this carefully selected set of relevant and optimal sub-trajectories as $\mathcal{D}(s, g^\star)$. During evaluation, we then dynamically adapt the policy to the current state-goal pair $(s, g^\star)$ by fine-tuning it on uniform samples from $\mathcal{D}(s, g^\star)$ for a few gradient steps, using the following objective:

$$J_{\text{TTT}}(\theta) = -\mathbb{E}_{s' \sim \mathcal{D}(s,g^\star)} \, \mathcal{L}(s', g^\star; \theta), \tag{5}$$

where we overload $\mathcal{D}$ to represent a uniform distribution over states in the dataset. Here, $\mathcal{L}$ is any standard policy

learning loss, such as behavior cloning or off-policy reinforcement learning.[2] While test-time training might use a different loss than pre-training, for simplicity, we use the same loss for TTT as for pre-training throughout. We set the goal for policy fine-tuning deterministically to the evaluation goal $g^\star$, as the policy will only be queried with this goal.

In the following, we discuss the two key components of GC-TTT: (1) selecting relevant and optimal experience from the dataset, and (2) fine-tuning the policy dynamically during evaluation.

### 4.1 *What to train on?* Selecting Relevant and Optimal Experience

The first step of GC-TTT is to select trajectories which are relevant to the current state of the agent and optimal for achieving the target goal. To determine the relevance of sub-trajectories in $\mathcal{D}$ to the agent's current state $s \in \mathcal{S}$, we leverage a notion of temporal distance. In practice, this can be estimated by the learned quasimetric $-V(s, g)$ of a value function estimate (Wang et al., 2023) or by the locally correct distance function $d$ conventionally exposed by the goal-conditioned reward function (Andrychowicz et al., 2017). We consider a sub-trajectory $(s_1, \dots) \in \mathcal{D}$ as related to the current state $s$ if $d(s, s_1) < \epsilon$ for some $\epsilon > 0$, normally also provided by the environment. This results in a filtered set of sub-trajectories of diverse lengths:

$$\textbf{Relevance: } \mathcal{D}_{\text{rel}}(s) = \{(s_1, \dots s_H) \in \mathcal{D} \,|\, d(s, s_1) < \epsilon\}. \tag{6}$$

The threshold $\epsilon$ may be selected adaptively such that $\mathcal{D}_{\text{rel}}(s)$ is of a desired size; however, in our evaluated environments, fixing $\epsilon$ to a constant was sufficient. We note that the distance function does not need to be globally accurate, but only locally.

While this operation selects sub-trajectories that are relevant to the agent's current state, not all of them might be useful for reaching the agent's target goal $g^\star \in \mathcal{G}$. We thus further filter the data to include only those sub-trajectories which are most likely to eventually reach $g^\star$. To measure this, we estimate the returns of the sub-trajectories if

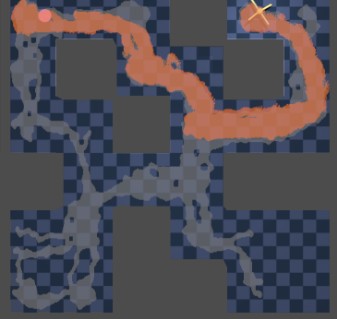

*Figure 2.* Visualization of data selection by GC-TTT in `antmaze play` during one evaluation episode (in orange). A random subset of trajectories from the dataset is shown in gray.

they were to be extended using the agent's current policy. We adopt an $H$-step return estimate (Sutton & Barto, 2018)

---

[2]For completeness, we include an overview in Appendix B.

which considers both the rewards along the sub-trajectory and the estimated value of its final state:

$$\widehat{V}((s_1, \ldots, s_H) | g^\star) = \sum_{i=1}^{H-1} \gamma^{i-1} R(s_i, g^\star) + \gamma^{H-1} V(s_H | g^\star). \quad (7)$$

In practice, the resulting estimate combines an evaluation of the behavioral policy inducing $(s_1, \ldots, s_H)$ with a value estimate of the current policy $\pi_\theta$. We remark that $H$ is not a hyperparameter, but rather the length of each sub-trajectory being considered. These return estimates rely on a value estimate (i.e., a critic), which is a core component across most offline GCRL algorithms. When a critic is not available, we can use simple trajectory returns according to the behavioral policy and the reward function $R$, as we demonstrate in Section 5. Given the return estimates $\widehat{V}(\tau \mid g^\star)$, we set the scalar $C$ to their $q$-th percentile among all relevant sub-trajectories in $\mathcal{D}_{\text{rel}}(s)$, and select the most optimal ones:

**Optimality:** $\mathcal{D}(s, g^\star) = \{\tau \in \mathcal{D}_{\text{rel}}(s) \mid \widehat{V}(\tau | g^\star) > C\}. \quad (8)$

### 4.2  *When to train?* Receding Horizon Training

It remains to decide when to fine-tune the policy based on the TTT objective from Equation (5). In principle, we can update the policy whenever either the agent's state or goal change. Since we expect neighboring states to lead to similar fine-tuned policies, we opt for a natural receding horizon approach (Morari & Lee, 1999). We describe the full GC-TTT algorithm in Algorithm 1 and provide a graphical example of its application in Appendix D.

Every $K$ steps, we re-initialize the agent to its pre-training weights [3]. Considering the current state $s$ and goal $g^\star$, we then fine-tune the pre-trained agent on relevant and optimal data. Crucially, the data selection is performed once, *prior* to test-time gradient steps, and thus relies on pre-trained estimators. We then roll out the fine-tuned policy for $K$ steps, before its weights are once again reset, and the entire process is repeated. Intuitively, each fine-tuning allows the agent to focus on actions to be taken in its immediate future. Crucially, this allows the policy to only focus on parts of its task, instead of trying to solve it all-at-once. Furthermore, this framework allows dynamic trajectory corrections during each rollout: if the agent strays away from the optimal trajectory, GC-TTT can select helpful data to correct the direction towards the final goal. From this perspective, there are clear parallels between this high-level routine, and model predictive control (MPC, Rawlings et al., 2017), though importantly, our approach does not require a model. We remark that the update rule of GC-TTT may also be applied in different ways than we present here. For

instance, it is possible to just fine-tune the policy once, e.g., at the start of the episode or when an error is detected.

### 4.3  Computational Efficiency

While GC-TTT leads to substantial performance gains, it incurs additional computational costs at test-time. This cost scales with several design choices; in particular, it scales linearly with the TTT frequency $1/K$ and with the number of gradient steps $N$ for each iteration. Each gradient update can be as efficient as two forward passes, of which one is required at each time step for standard evaluation. Moreover, there is an overhead at each fine-tuning iteration due to data selection: if parallelization is possible (e.g., on graphics accelerators), this can be near-constant in practice, otherwise the overhead increases linearly with the number of samples $|\mathcal{D}|$. Finally, this cost is not distributed evenly through the evaluation, but rather at regular intervals, which can result in a non-constant control frequency. In practice, we find that

---

**Algorithm 1** Goal-conditioned Test-time Training

**Require:** Pre-trained policy parameters $\theta$, dataset $\mathcal{D}$, horizon $K$, number of gradient steps $N$, learning rate $\alpha$, distance $d$, goal-conditioned value estimate $\widehat{V}$, locality threshold $\epsilon$, percentile $q$.
1: **for** each evaluation episode **do**
2:     $s \sim \mu_0, g^\star \sim \mu_g$
          $\mapsto$ sample initial state and evaluation goal
3:     $\bar{\theta} \leftarrow \theta$  $\mapsto$ store policy parameters
4:     **while** not done **do**
5:         $\mathcal{D}_{\text{rel}}(s) \leftarrow \{(s_1, \ldots) \in \mathcal{D} \mid d(s, s_1) < \epsilon\}$
              $\mapsto$ select **relevant** sub-trajectories (Eq. 6)
6:         $C \leftarrow q$-th percentile of $\{\widehat{V}(\tau | g^\star) \mid \tau \in \mathcal{D}_{\text{rel}}(s)\}$
7:         $\mathcal{D}(s, g^\star) \leftarrow \{\tau \in \mathcal{D}_{\text{rel}}(s) \mid \widehat{V}(\tau | g^\star) \geq C\}$
              $\mapsto$ filter to **optimal** sub-trajectories (Eq. 8)
8:         **for** $i \in [1, \ldots, N]$ **do**
9:             $\theta \leftarrow \theta - \alpha \nabla_\theta \mathbb{E}_{s' \sim \mathcal{D}(s, g^\star)} \mathcal{L}(s', g^\star; \theta)$
                  $\mapsto$ fine-tune policy locally (Eq. 5)
10:        **end for**
11:        **for** $i \in [1, \ldots, K]$ **do**
12:            $a \sim \pi_\theta(s \mid g)$   $\mapsto$ sample action
13:            $s \sim P(\cdot \mid s, a)$   $\mapsto$ execute action
14:        **end for**
15:        $\theta \leftarrow \bar{\theta}$  $\mapsto$ reset policy
16:    **end while**
17: **end for**

---

GC-TTT (with a generous compute budget as described in Appendix F.3) completes a single evaluation episode (1000 steps) in $\sim 70$ seconds, for an average control frequency $> 10$ Hz [4]. While performance can be further improved by

---

[3]We remark that GC-TTT is a purely offline method, and is not designed for continual improvement: as such, it performs periodic resets to guarantee a stable initialization for test-time training.

[4]A significant share of compute is allocated at the start of the episode in order to prepare return estimates, and the rest is concentrated on TTT iterations, as detailed in Appendix F.3.

efficient implementations and more performant hardware, this number is comparable to the inference speed of methods relying on efficient model-based planning (Pinneri et al. (2020), ∼12-20Hz), or VLAs with diffusion heads (Black et al. (2024), ∼5-15Hz). For context, a critic-free version of the algorithm with less frequent TTT and the pre-trained policy reach a control frequency of > 75 and ∼ 190 Hz, respectively. For an empirical study of the trade-off between performance and compute requirements, we refer to Section 5.

## 5 Experiments

We provide an empirical validation of our contributions spanning four environments and three algorithmic backbones. We identify five main insights, which we present in the following. Our code is available at the anonymous website. We refer to Appendix C for additional experimental results and to Appendix F for implementation details.

**Environments** We rely on a suite of goal-conditioned tasks from OGBench (Park et al., 2025). Namely, we evaluate three loco-navigation tasks of increasing complexity (`pointmaze`, `antmaze` and `humanoidmaze`), spanning from 2 to 21 degrees of freedom.

We evaluate all environments in their `medium` instance, across two datasets of different qualities, namely `navigate` and `stitch`. The former includes full demonstrations for any evaluation state-goal pair, while the latter may only be solved by "stitching" different trajectories together. For ease of interpretation, we refer to them as `expert` and `play`, respectively. We additionally consider one manipulation task, in which a robotic arm is tasked with relocating a cube (`cubesingle`).

**Backbones** In principle, GC-TTT is applicable across the broad class of value-based offline goal-conditioned algorithms. We thus select a representative subset of algorithms, and focus our evaluation on GC-BC (Yang et al., 2022), GC-IQL (Kostrikov et al., 2022) and SAW (Zhou & Kao, 2025). GC-BC (behavior cloning) is a supervised algorithm for goal-conditional imitation, which directly matches the policy's output to the actions present in the offline dataset. GC-IQL is an implicit method for offline RL, which bypasses evaluation on out-of-distribution actions through expectile regression. We evaluate it in combination with two policy extraction objectives: DDPG+BC (Fujimoto & Gu, 2021) and AWR (Peng et al., 2019), the latter of which is reported in Appendix C.6. SAW (Zhou & Kao, 2025) can be seen as a hierarchical offline reinforcement learning algorithm, which directly amortizes high-level planning in the low-level policy. We select BC and IQL due to their widespread adoption, and their representativeness of on-policy and off-policy learning in offline settings, respectively. With SAW as a backbone, GC-TTT achieves state-of-the-art performance on the considered benchmarks.

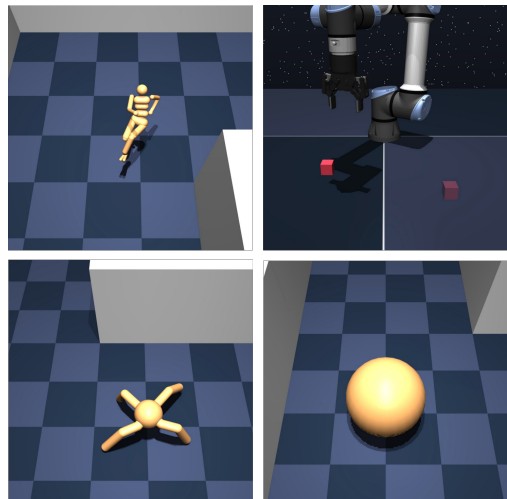

*Figure 3.* The four environments from OGBench (Park et al., 2025): from top left in clockwise order, `humanoidmaze`, `cubesingle`, `antmaze`, `pointmaze`.

**Insight 1: GC-TTT substantially improves the policy across diverse environments and learning algorithms.** To begin with, we evaluate the performance of GC-TTT across the described array of environments and algorithms. We train the backbone algorithm until convergence and report the average performance at 800k, 900k and 1M gradient steps, as in the protocol described by (Park et al., 2025). Performances are computed as the average success rate across four fixed goals in each environment; we report mean and standard error across 3 seeds. We report our results in Table 1 and Figure 4. We observe that GC-TTT improves the performance of the backbone for the majority of backbone-environment combinations, and does not impact it negatively in the remaining ones. Interestingly, test-time training is capable of reliably solving `pointmaze` with simple techniques (i.e., GC-BC). This suggests that standard approaches for offline goal-conditioned RL might fail to retrieve a performant multi-goal policy (e.g., due to training dynamics), but can be quickly adapted to perform well on a single goal, as a few gradient steps are sufficient to significantly improve their policies. We note that the mapping that GC-TTT seeks at test time is from a *single* goal and a *subset* of the state space to optimal actions, and thus arguably easier to fit compared to an optimal policy for all states and goals. This sheds some light on one of the open problems discussed in Park et al. (2025). Furthermore, as the environment complexity increases (e.g., `antmaze` or `humanoidmaze`), the improvements induced by GC-TTT remain significant; and `cubesingle` confirms that this trend holds in settings with fundamentally different dynamics.

**Insight 2: GC-TTT can be applied without value estimates if expert data is available.** We now turn our attention to a critic-free variant of GC-TTT. This algorithm

| | pointmaze | | antmaze | | humanoidmaze | | cubesingle | **avg.** |
|---|---|---|---|---|---|---|---|---|
| | expert | play | expert | play | expert | play | | |
| GC-BC | 0.05 (0.01) | 0.30 (0.09) | 0.29 (0.02) | 0.47 (0.05) | 0.07 (0.02) | 0.33 (0.01) | **0.05** (0.02) | 0.22 |
| + TTT (no critic) | **0.76** (0.04) | – | **0.54** (0.04) | – | 0.15 (0.02) | – | – | – |
| + TTT | **0.78** (0.02) | **0.93** (0.02) | **0.48** (0.03) | **0.76** (0.03) | **0.21** (0.01) | **0.68** (0.03) | 0.08 (0.01) | 0.56 |
| GC-IQL-DDPG | 0.50 (0.03) | 0.21 (0.04) | 0.64 (0.05) | 0.24 (0.05) | 0.32 (0.03) | 0.10 (0.02) | **0.74** (0.03) | 0.39 |
| + TTT (no critic) | 0.55 (0.02) | – | **0.80** (0.03) | – | 0.40 (0.02) | – | – | – |
| + TTT | **0.60** (0.00) | **0.38** (0.05) | **0.85** (0.02) | **0.68** (0.05) | **0.56** (0.03) | **0.49** (0.03) | 0.75 (0.02) | 0.61 |
| SAW | 0.96 (0.01) | 0.84 (0.03) | **0.97** (0.01) | **0.98** (0.01) | 0.85 (0.03) | 0.78 (0.02) | **0.73** (0.03) | 0.87 |
| + TTT (no critic) | **0.99** (0.01) | – | **0.97** (0.02) | – | 0.87 (0.04) | – | – | – |
| + TTT | **0.99** (0.01) | **0.97** (0.01) | **0.97** (0.01) | **0.97** (0.01) | **0.92** (0.03) | **0.84** (0.01) | 0.71 (0.03) | 0.91 |

*Table 1.* Success rates of GC-TTT and its critic-free variant across loco-navigation and manipulation, on top of GC-BC, GC-IQL-*DDPG*, and SAW. Numbers in parentheses are standard errors across 5 seeds. **Bold** numbers denote results that are within the standard error of the best for a given backbone. Underlined numbers denote whether significantly TTT outperforms pre-training. The hyperparameters of GC-TTT are tuned per environment (cf. Appendix F.1). We report results for fixed hyperparameters in Table 4.

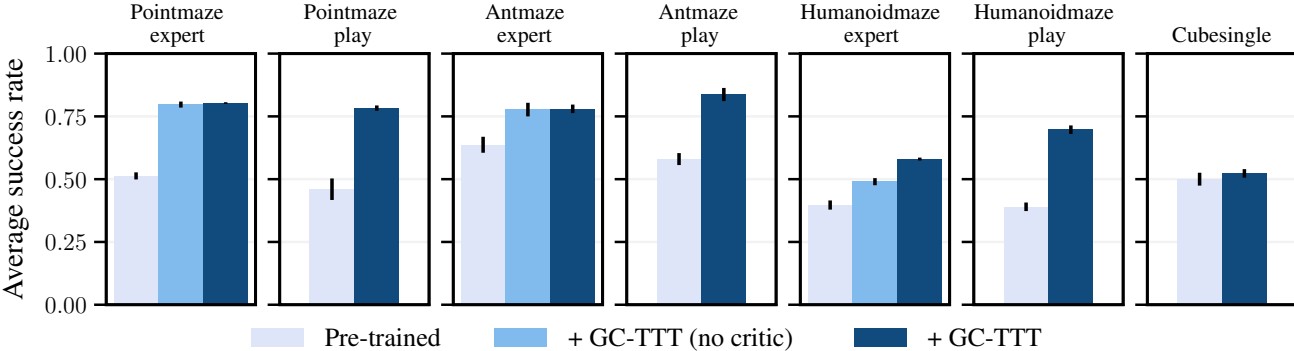

*Figure 4.* Success rates of GC-TTT within each environment, averaged across RL backbones.

replaces the $H$-step return estimate (cf. Equation (7)) with the trajectory returns (i.e., a discounted sum of rewards along the trajectory). As such, this variant does not require additionally training a critic network (and thus combines seamlessly with, e.g., BC). However, this critic-free variant cannot infer optimality from trajectories that do not reach the target goal, and is therefore limited to expert data. As shown in Table 1 and Figure 4, on such tasks with expert data, the critic-free variant retains much of the effectiveness of GC-TTT. In contrast, in play tasks, all relevant sub-trajectories are likely to achieve the same trajectory return of 0.

**Insight 3: Selecting both relevant and optimal data is necessary.** A core component of GC-TTT is the selection of *relevant* and *optimal* data from the offline dataset (cf. Section 4.1). We ablate this design choice in Figure 5 (left), where we report the average success rates with GC-IQL as backbone in the pointmaze/antmaze play environments. We observe that selecting random data from the dataset is not effective, as the global objective of the backbone algorithm has already converged. Selecting relevant but suboptimal data marginally improves perfor-

mance, possibly encouraging a form of test-time behavior regularization. Selecting optimal data that may be irrelevant to the agent's current state yields a slight increase in success rate. We attribute this to the relatively small size of the environments, which means that by chance some selected trajectories might also be relevant. Remarkably, GC-TTT leads to a substantial performance gain by combining both relevance to the agent's current state and optimality for the agent's goal. We additionally plot data selected by GC-TTT over the course of an evaluation episode in Figure 2.

**Insight 4: The frequency of test-time training should adapt depending on the difficulty of the environment.** The compute cost of GC-TTT scales linearly in the frequency of test-time training. Hence, from this perspective, updating the policy less frequently seems desirable. At the same time, frequent updates allow the agent to focus on local information and to quickly correct when diverging from the optimal path to the goal. We demonstrate this in Figure 5 (middle), where we evaluate GC-TTT with GC-IQL in antmaze play while keeping a fixed number of gradient steps per iteration ($N = 200$). We find that the

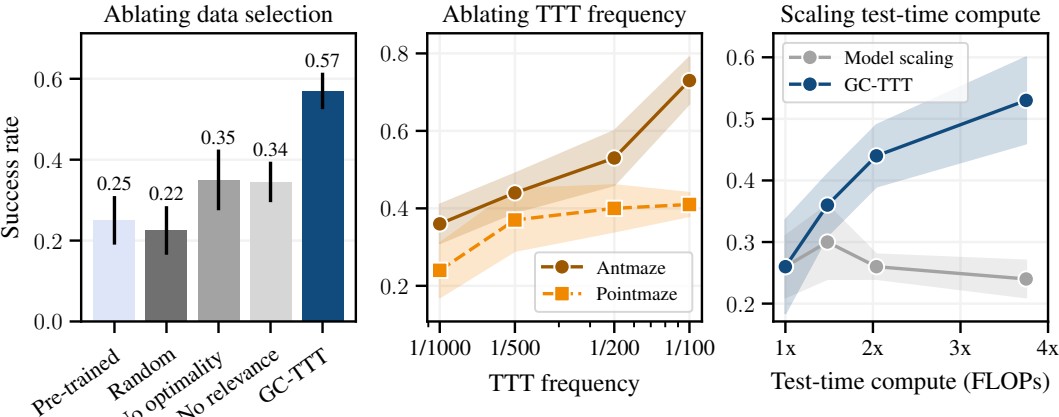

*Figure 5.* **Left:** Ablation of the data selection criteria. Both relevance and optimality have to be considered to filter the dataset for test-time training. **Middle:** Allocating more compute by increasing the frequency of TTT improves performance, and saturates slightly earlier in simpler environments. **Right:** We compare scaling test-time compute of GC-TTT (by increasing TTT frequency) to scaling the policy networks such that inference FLOPs are matched, within the `antmaze play` environment. We find that GC-TTT scales well with increased test-time compute, while scaling model size does not yield significant improvements.

value estimates used for data selection are not accurate over long horizons ($> 200$ steps in `antmaze play`), leading to poor performance if the policy is updated too infrequently.[5] However, as the frequency of TTT increases, we observe that GC-TTT leads to significant performance gains. We repeat the same experiment on `pointmaze play`, which is an arguably simpler environment. We observe that performance already saturates at a lower frequency (i.e., $1/200$), suggesting that test-time training should be applied at shorter intervals in more complex environments.

**Insight 5: GC-TTT scales better than model size.** Having shown that GC-TTT predictably improves when allocating more compute, we analyze another option to scale test-time compute, namely by training larger policies, which are more expensive to evaluate. For this, we compare the performance of GC-TTT with a given frequency $1/K$ to the performance of larger policies that are not trained at test-time, but which have matched inference FLOPs to GC-TTT. To match the inference FLOPs of GC-TTT scaling and model scaling, we assume that compute requirements scale linearly with TTT frequency, but quadratically in the width of the policy. Details on how compute costs are calculated can be found in Appendix F. In Figure 5 (right), we find that GC-TTT consistently outperforms model scaling across a broad range of inference FLOPs.

## 6   Conclusion and Future Work

This work introduces a framework of test-time training for offline goal-conditioned RL. We propose a self-supervised

data selection scheme which chooses relevant and optimal data for the agent's current state and goal from an offline dataset of trajectories. Our proposed method, GC-TTT, periodically fine-tunes the pre-trained policy on this data during evaluation. We find that GC-TTT consistently leads to significant improvements across several environments and underlying RL algorithms.

The main practical limitation of this work arguably lies in its compute requirements, which we discuss in Section 4.3. While our measured average control frequency of GC-TTT is compatible with some robotic applications, high-frequency control would require addressing or distributing episodic delays, possibly through the development of a lazy variant of GC-TTT, or by decreasing the test-time training frequency. While this was not observed in the benchmark we considered, gradient-based policy updates may also potentially lead to instability: constraining the magnitude of the updates would be crucial in safety-critical systems. Finally, GC-TTT relies on reasonable value estimates and on available data related to the agent's current state and goal: it may thus fail to provide substantial improvements when data is scarce, or the state/goal space is immense, e.g. in complex, multi-object manipulation scenarios.

By showing that test-time training can effectively improve policies from off-policy experiential data, our work opens up several exciting directions for further research. On a practical level, our findings suggest that current offline GCRL algorithms are unable to accurately fit each of the tasks they are trained on. The reason for this should be investigated, and might suggest directions for improving offline RL pre-training. Moreover, GC-TTT does not leverage the data that is freshly collected at test-time, beyond the current state. We believe that leveraging this

---

[5]An alternative to dynamically retraining during evaluation, which leads to a constant control frequency, but we do not evaluate here, is to reset the policy after $K$ steps to the pre-trained policy.

new experience with a test-time online RL algorithm is an exciting direction. Finally, the framework proposed in this work can be readily extended beyond goal-reaching tasks to more general decision-making settings, including other domains such as reasoning in natural language. We expect that progressively shifting computational resources to test-time training can substantially improve performance in areas ranging from robotic control to reasoning agents.

## Acknowledgements

Marco Bagatella and Mert Albaba are supported by the Max Planck ETH Center for Learning Systems. Georg Martius is a member of the Machine Learning Cluster of Excellence, EXC number 2064/1 –Project number 390727645. We acknowledge the support from the German Federal Ministry of Education and Re-search (BMBF) through the Tübingen AI Center (FKZ:01IS18039B), from the European Research Council (ERC) under the European Union's Horizon 2020 research and Innovation Program Grant agreement no. 815943, and from the Swiss National Science Foundation under NCCR Automation, grant agreement 51NF40 180545.

## Impact Statement

This work presents a general framework for fine-tuning control policies. As such, we believe that its potential impacts no not differ significantly from the general concerns surrounding research in deep reinforcement learning (e.g., automation, misuse of hardware).

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

# A    Taxonomy of Test-time Training

Test-time training (TTT) describes a family of methods that update model parameters at test-time for each task. We categorize various approaches to TTT below.

| Category | Methods |
|----------|---------|
| Imitating expert data | Often referred to as "Test-Time Training" (TTT), e.g., (Hardt & Sun, 2024; Hübotter et al., 2025; Akyürek et al., 2025) |
| Learning from any experience | Test-Time Offline Reinforcement Learning (TTORL) |
| Learning from self-generated experience | Test-Time (Online) Reinforcement Learning (TTRL), e.g., (Zuo et al., 2025; Diaz-Bone et al., 2025) |

*Table 2.* Taxonomy of test-time training.

# B    Discussion of Offline RL Algorithms

The empirical validation of this work builds upon three widespread algorithms for extracting policies from offline data. In this section, we provide a concise introduction to them.

## B.1    Behavior Cloning

Behavior Cloning (Ross et al., 2011) is a standard approach for policy learning, which reduces a control problem to supervised reconstruction. Given a distribution $\mu$ over state-action pairs, a policy $\pi_\theta$ is trained by minimizing

$$\mathcal{L}_{\text{BC}}(\theta) = -\mathbb{E}_{(s,a)\sim\mu} \log \pi_\theta(a \mid s). \tag{9}$$

The resulting policy will maximize the likelihood of actions in the dataset, and thus converge to the behavioral policy, if it belongs to the policy class.

## B.2    Implicit Q-Learning

Implicit Q-Learning (Kostrikov et al., 2022) is an offline RL algorithm, which avoids querying the critic on out-of-distribution actions, and directly estimates a value function through expectile regression. Given a distribution $\mu$ of state-action-next state transitions labeled with a reward, IQL defines the following losses:

$$\mathcal{L}_Q(\phi) = \mathbb{E}_{(s,a,r,s')\sim\mu} \left(r + \gamma V_\psi(s') - Q_\phi(s,a)\right)^2, \tag{10}$$

and

$$\mathcal{L}_V(\psi) = \mathbb{E}_{(s,a,r)\sim\mu} L^\alpha(Q_\phi(s,a) - V_\psi(s)) \quad \text{with } L^\alpha(x) = |\alpha - \mathbf{1}_{x<0}|x^2. \tag{11}$$

As the expectile $\alpha$ approaches one, $V$ approximates the maximum of $Q$. Thus, IQL is capable of off-policy learning, and can estimate the value function of the optimal policy (Kostrikov et al., 2022). An optimal policy may then be extracted through advantage weighted regression (AWR, Peng et al., 2019):

$$\mathcal{L}_\pi(\theta) = -\mathbb{E}_{(s,a,r)\sim\mu} \exp\left(\beta\big(Q_\phi(s,a) - V_\psi(s)\big)\right) \log \pi_\theta(a \mid s), \tag{12}$$

where $\beta$ interpolates between extracting the behavior policy or the greedy one. Alternatively, a policy can also be estimated through a BC-regularized, DDPG-style loss (Fujimoto & Gu, 2021):

$$\mathcal{L}_\pi(\theta) = -\mathbb{E}_{(s,a,r)\sim\mu} \beta Q_\phi(s,\hat{a}) + \log \pi_\theta(a \mid s) \quad \text{with } \hat{a} \sim \pi_\theta(s). \tag{13}$$

## B.3    SAW

SAW (Zhou & Kao, 2025) is an offline reinforcement learning algorithm designed to flatten hierarchical approaches (Park et al., 2023).

At its core, it relies on implicit Q-learning for estimating a value function, and on AWR for policy extraction, with an additional term encouraging alignment of the low-level policies across close and distant goals:

$$\mathcal{L}_{\text{SAW}}(\theta) = -\mathbb{E}_{(s,a,r)\sim\mu} \exp\left(\alpha(V(w,g) - V(s,g))\right) D_{\text{KL}}(\pi_\theta(\cdot \mid s,g)\|\pi_\psi(\cdot \mid s,w)), \tag{14}$$

where we made the dependencies of the value functions and policies on goals explicit, $w$ is a candidate subgoal, and $\pi_\psi$ is simply trained with AWR.

## C  Additional Experiments

### C.1  Ablation on Test-time Training Parameters

Figure 6 presents the success rate of GC-TTT with GC-IQL on `antmaze` play as the number of test-time training gradient steps $N$ changes. We observe that increasing the number of gradient steps helps initially, as the policy can better fit the local data. However, an excessive number of gradient steps may decrease performance, as the policy is trained on a small dataset, and offline issues such as value overestimation may arise. Regarding the learning rates, the higher learning rate facilitates quicker adaptation and shows a slight advantage in peak performance. While there are differences, both learning rates yield comparable results as gradient steps increases.

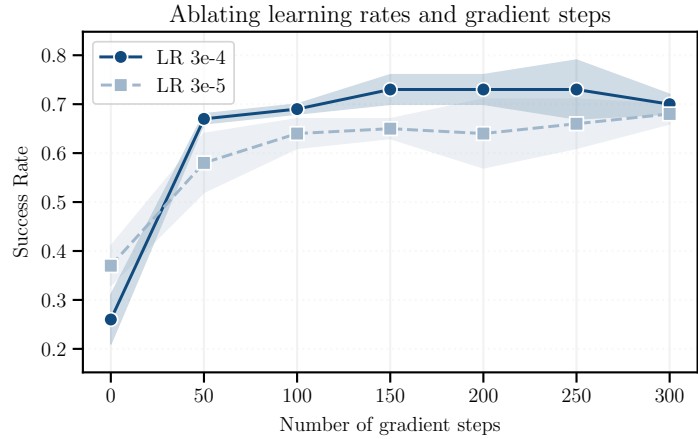

*Figure 6.* GC-TTT results for different gradient steps.

### C.2  Ablation on Data Selection Parameters

The data selection criteria (Equations 6 and 8) introduce a few parameters: the relevance threshold $\epsilon$, the horizon for $H$-step returns, and the optimality quantile $q$. This section discusses each of them, and validates their impact. We perform each ablation on top of GC-IQL in `antmaze` with play-like data.

The relevance threshold $\epsilon$ is not treated as a hyperparameter. While precise tuning is likely to improve performance, we directly adopt the value specified by the reward function which the environment defines. For instance, in the case of `antmaze`, the distance function $d$ is simply the L2 distance between the robot's center of mass coordinate and the goal, and $\epsilon$ is set to 1. Figure 7 (left) reports the effect of $\epsilon$ on performance if it was tuned: as expected, extreme values degrade performance, as they induce scarcity in selected data, or select irrelevant sub-trajectories.

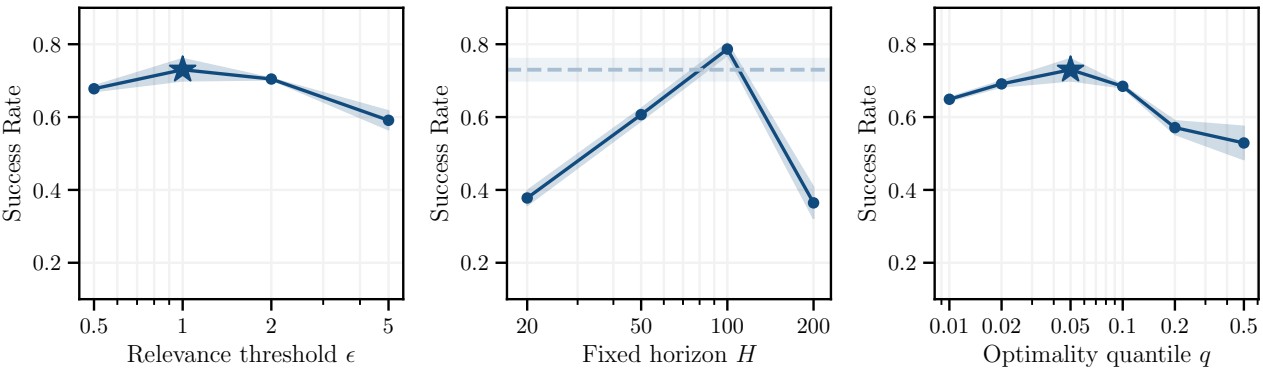

*Figure 7.* Ablation of parameters controlling data selection: from left to right, relevance threshold $\epsilon$, return horizon $H$ and optimality quantile $q$.

|  | Reward-based | Value-based (C=-14) | Value-based (C=-18) | Value-based (C=-22) |
|---|---|---|---|---|
| `antmaze` play | 0.73 (0.01) | 0.68 (0.04) | 0.73 (0.01) | 0.67 (0.03) |

*Table 3.* Success rates of GC-TTT with the original relevance criterion and a value-based version, on top of GC-IQL. Numbers in parentheses are standard errors across 3 seeds.

The return horizon $H$ is also not a hyperparameter: it is always set to the length of each considered sub-trajectory (or, in other words, it is adapted dynamically). The goal of the optimality criterion is to evaluate the expected return of a policy which initially follows each sub-trajectory. As such, MC and $H$-step returns both provide reasonable estimates: the former assumes that the behavioral policy is followed after the trajectory, and the latter assumes that the learned policy is followed instead. For completeness, we provide a comparison of the current strategy with one that evaluates each sub-trajectory through $H$-step returns with a fixed $H$. In Figure 7 (center) we observe that adapting $H$ dynamically (represented by a dashed horizontal line) matches the performance of hand-selecting a fixed horizon.

The optimality quantile $q$ is also kept fixed across our experiments, but, unlike the previous two, it has no fundamental connections to other quantities. We thus provide an ablation of this hyperparameter in Figure 7 (right). We observe that, while GC-TTT is rather robust to this choice, however, extremely low and high values may lower performance, as they result in an overly strict or lax optimality criterion, respectively.

### C.3   Value-based Relevance Criterion

The relevance criterion defined in Equation 6 relies on the reward criterion normally exposed in goal-conditioned settings. When this is not available, however, the criterion may be replaced by a proxy based on a value estimate:

$$\textbf{Value-based relevance:} \qquad \mathcal{D}_{\mathrm{rel}}(s) = \{(s_1, \dots s_H) \in \mathcal{D} \mid V(s, s_1) > C\}. \qquad (15)$$

This time, $C$ is a constant hyperparameter, which, similarly to $\epsilon$, can control the maximum temporal distance between the current state $s$ and selected trajectories.

We find that, empirically, this modification does not affect performance significantly: we report performance of GC-TTT with the original and the value-based relvance criterion in `antmaze` in Table 3.

### C.4   Parameter Scaling Ablation

Figure 5 (right) studies the extent to which performance may be improved by scaling the parameter count of the policy. In order to ensure that the absence of improvement does not stem from hyperparameter choices, we additionally report results for different learning rates in Figure 8.

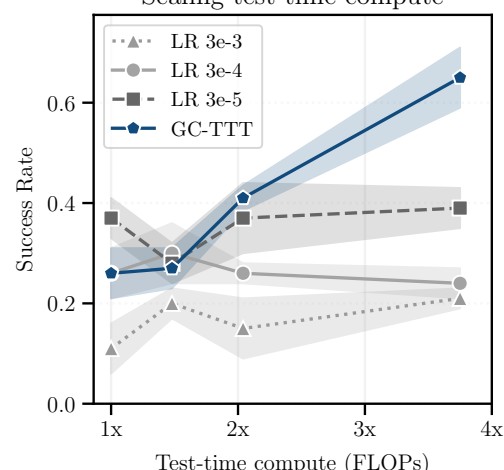

*Figure 8.* Model scaling results for different learning rates.

### C.5   Results for Fixed Hyperparameters

This section reports results for the same evaluation as in Table 1, but fixes the same test-time hyperparameters across all environments. Table 4 suggests that the drop in performance due to environment-agnostic tuning is moderate. We further evaluate the same setting while fixing all hyperparameters across both environments and algorithmic backbones in Table 5, and observe that average performance does not drop significantly further.

### C.6   Results for GC-IQL-AWR

We additionally report results for GC-TTT on GC-IQL, when replacing the policy extraction objective with AWR. For convenience, this evaluation is reported in Table 4.

| | pointmaze | | antmaze | | humanoidmaze | | cubesingle | **avg.** |
|---|---|---|---|---|---|---|---|---|
| | expert | play | expert | play | expert | play | | |
| GC-BC | 0.05 (0.01) | 0.33 (0.10) | 0.30 (0.03) | 0.50 (0.05) | 0.06 (0.01) | 0.33 (0.01) | **0.07** (0.03) | 0.23 |
| + TTT (no critic) | **0.78** (0.02) | – | **0.49** (0.02) | – | 0.09 (0.00) | – | – | – |
| + TTT | **0.79** (0.04) | **0.96** (0.01) | **0.48** (0.04) | **0.75** (0.02) | **0.19** (0.01) | **0.64** (0.02) | 0.09 (0.01) | 0.55 |
| GC-IQL-AWR | 0.16 (0.05) | 0.31 (0.07) | 0.58 (0.06) | 0.26 (0.05) | 0.11 (0.04) | 0.03 (0.02) | 0.57 (0.02) | 0.28 |
| + TTT (no critic) | 0.67 (0.03) | – | **0.71** (0.04) | – | 0.19 (0.03) | – | – | – |
| + TTT | **0.79** (0.02) | **0.81** (0.03) | **0.65** (0.09) | **0.64** (0.04) | **0.22** (0.04) | 0.19 (0.02) | **0.61** (0.05) | 0.55 |
| GC-IQL-DDPG | 0.52 (0.04) | 0.24 (0.07) | 0.65 (0.09) | 0.26 (0.05) | 0.27 (0.02) | 0.08 (0.03) | **0.72** (0.05) | 0.39 |
| + TTT (no critic) | 0.53 (0.07) | – | 0.76 (0.02) | – | 0.39 (0.03) | – | – | – |
| + TTT | **0.58** (0.02) | **0.38** (0.04) | **0.81** (0.02) | **0.73** (0.03) | **0.49** (0.04) | **0.46** (0.02) | 0.65 (0.06) | 0.58 |
| SAW | 0.97 (0.01) | 0.81 (0.04) | 0.96 (0.01) | **0.98** (0.01) | 0.86 (0.05) | 0.76 (0.04) | **0.71** (0.05) | 0.86 |
| + TTT (no critic) | **1.00** (0.00) | – | **0.95** (0.03) | – | 0.86 (0.04) | – | – | – |
| + TTT | **0.99** (0.01) | **0.92** (0.03) | **0.99** (0.01) | 0.93 (0.03) | **0.93** (0.02) | **0.79** (0.06) | 0.73 (0.04) | 0.89 |

*Table 4.* Success rates of GC-TTT and its critic-free variant across loco-navigation and manipulation, on top of GC-BC, GC-IQL-*DDPG*, and SAW. Numbers in parentheses are standard errors across 3 seeds. **Bold** numbers denote results that are within the standard error of the best for a given backbone. Underlined numbers denote whether TTT outperforms pre-training. The hyperparameters of GC-TTT are fixed across environments.

| | pointmaze | | antmaze | | humanoidmaze | | cubesingle | **avg.** |
|---|---|---|---|---|---|---|---|---|
| | expert | play | expert | play | expert | play | | |
| GC-BC | 0.05 (0.01) | 0.33 (0.10) | 0.30 (0.03) | 0.50 (0.05) | 0.06 (0.01) | 0.33 (0.01) | **0.07** (0.03) | 0.23 |
| + TTT (no critic) | **0.77** (0.03) | – | **0.54** (0.07) | – | **0.13** (0.03) | – | – | – |
| + TTT | **0.79** (0.04) | **0.94** (0.03) | **0.48** (0.04) | **0.74** (0.01) | **0.14** (0.04) | **0.71** (0.03) | 0.09 (0.01) | 0.55 |
| GC-IQL-DDPG | 0.52 (0.04) | **0.24** (0.07) | 0.65 (0.09) | 0.26 (0.05) | 0.27 (0.02) | 0.08 (0.03) | **0.72** (0.05) | 0.39 |
| + TTT (no critic) | **0.58** (0.01) | – | **0.80** (0.05) | – | 0.31 (0.04) | – | – | – |
| + TTT | **0.58** (0.02) | 0.36 (0.08) | **0.85** (0.01) | 0.69 (0.01) | **0.42** (0.03) | **0.46** (0.03) | 0.74 (0.04) | 0.59 |
| SAW | 0.97 (0.01) | 0.81 (0.04) | **0.96** (0.01) | **0.98** (0.01) | 0.86 (0.05) | 0.76 (0.04) | **0.71** (0.05) | 0.86 |
| + TTT (no critic) | **0.98** (0.02) | – | **0.96** (0.01) | – | **0.86** (0.02) | – | – | – |
| + TTT | **0.98** (0.01) | **0.92** (0.02) | **0.96** (0.01) | **0.95** (0.02) | 0.90 (0.03) | **0.84** (0.01) | 0.69 (0.08) | 0.89 |

*Table 5.* Success rates of GC-TTT and its critic-free variant across loco-navigation and manipulation, on top of GC-BC, GC-IQL-*DDPG*, and SAW. Numbers in parentheses are standard errors across 3 seeds. **Bold** numbers denote results that are within the standard error of the best for a given backbone. Underlined numbers denote whether TTT outperforms pre-training. The hyperparameters of GC-TTT are fixed across environments and algorithmic backbones.

| | pointmaze | | antmaze | | humanoidmaze | | cubesingle | **avg.** |
|---|---|---|---|---|---|---|---|---|
| | expert | play | expert | play | expert | play | | |
| QRL | **0.91** (0.04) | 0.72 (0.05) | **0.76** (0.03) | 0.62 (0.05) | **0.09** (0.05) | 0.18 (0.01) | 0.02 (0.01) | 0.47 |
| + TTT (no critic) | **0.92** (0.02) | – | **0.77** (0.03) | – | **0.09** (0.03) | – | – | – |
| + TTT | **0.91** (0.03) | **0.83** (0.03) | **0.79** (0.03) | **0.74** (0.01) | **0.14** (0.02) | **0.26** (0.01) | **0.05** (0.01) | 0.53 |

*Table 6.* Success rates of GC-TTT and its critic-free variant across loco-navigation and manipulation, on top of QRL. Numbers in parentheses are standard errors across 3 seeds. **Bold** numbers denote results that are within the standard error of the best for a given backbone. Underlined numbers denote whether TTT significantly outperforms pre-training. The hyperparameters of GC-TTT are fixed across environments.

| | $\alpha = 0.0$ | $\alpha = 0.1$ | $\alpha = 1.0$ |
|---|---|---|---|
| antmaze play | 0.73 (0.06) | 0.78 (0.03) | 0.31 (0.02) |

*Table 7.* Success rates of GC-TTT with noisy value estimates, on top of GC-IQL. Numbers in parentheses are standard errors across 3 seeds.

## C.7  Results for QRL

Our evaluation main evaluation in Table 1 revolves on GCBC, GCIQL and SAW as diverse and representative baselines for offline GCRL. QRL (Wang et al., 2023) is a strong method which relies on quasimetric networks and a principled objective to retrieve optimal goal-reaching policies. We include results for GC-TTT on top of QRL across our benchmark in Table 6, using a learning rate $\alpha = 3e^{-4}$, $N = 200$ gradient steps, an horizon of $K = 500$, and mixing selected data with uniform samples with a 50% ratio. While QRL already achieved strong performance on expert dataset, we find that GC-TTT consistently improves performance from play-like datasets, further supporting the algorithmic-agnostic nature of GC-TTT.

## C.8  Robustness to Noisy Value Estimates

While GC-TTT relies on an estimated value function for data selection, our experiments suggest that it is rather robust to inaccuracies. In particular, we find that ranking subtrajectories does not require the same degree of accuracy that policy extraction does (see Figure 9). To provide further evidence, we additionally report the performance of GC-TTT in antmaze play as noise is injected into the value estimates for optimality-based data selection (Eq. 7, 8) in Table 7. Upon computing value estimates (Eq. 7), we compute the average absolute value $\bar{V}$ across a batch, and corrupt each value estimate with Gaussian noise with standard deviation $\sigma = \alpha\bar{V}$. When increasing $\alpha$, we observe that performance does not degrade significantly for modest perturbations, and only does so under strong perturbations ($\alpha = 1$).

## C.9  Statistical Tests

Table 1 marks improvement as significant when the confidence intervals denoted by standard errors do not overlap. We additionally run a Welch's t-test as reported in Table 8, confirming that performance improvements are significant ($p < 0.1$) in 14/21 environment/backbone combinations (most non-significant improvements occur with SAW as a backbone, which already masters some of the tasks).

## D  Visualizing GC-TTT

As the test-time training process may be divided in two parts (data selection and agent update), it remains rather interpretable and, in low-dimensional environments, easy to visualize. We thus provide a visualization of the GC-TTT on pointmaze with play-like data in Figure 9. In this case, two iterations are sufficient to solve this task, each of whom is displayed in a row. Let us focus on the first iteration. As actions are 2-dimensional, we begin by plotting the base policy (pre-trained until convergence with GC-IQL). As highlighted by the red circle, the actions predicted in the initial state (denoted by a diamond) for the current goal (denoted by a star) are wrongly pointing to the bottom left. GC-TTT selects sub-trajectories that go around the obstacle (as seen in the second column), and after test-time-training on them, the policy correctly predicts actions going right (third column, in the red circle). The final column displays the test-time training objective (total loss of GC-IQL, in this case), which steadily decreases in a handful of gradient steps. After 200 steps, TTT is applied again, resulting in a policy that completes the final part of the task.

| Environment | Base SR | SR w/ TTT | $t$ | $p$ (sig.) |
|---|---|---|---|---|
| GC-BC vs GC-BC + TTT | | | | |
| `pointmaze` expert | 0.05 0.01) | 0.78 (0.02) | 32.65 | 0.000 (***) |
| `pointmaze` play | 0.30 (0.09) | 0.93 (0.02) | 6.83 | 0.016 (**) |
| `antmaze` expert | 0.29 (0.02) | 0.48 (0.03) | 5.27 | 0.009 (***) |
| `antmaze` play | 0.47 (0.05) | 0.76 (0.03) | 4.97 | 0.013 (**) |
| `humanoidmaze` expert | 0.07 (0.02) | 0.21 (0.01) | 6.26 | 0.009 (***) |
| `humanoidmaze` play | 0.33 (0.01) | 0.68 (0.03) | 11.07 | 0.004 (***) |
| `cubesingle` | 0.05 (0.02) | 0.08 (0.01) | 1.34 | 0.274 (ns) |
| GC-IQL-DDPG vs GC-IQL-DDPG + TTT | | | | |
| `pointmaze` expert | 0.50 (0.03) | 0.60 (0.00) | 3.33 | 0.079 (*) |
| `pointmaze` play | 0.21 (0.04) | 0.38 (0.05) | 2.65 | 0.060 (*) |
| `antmaze` expert | 0.64 (0.05) | 0.85 (0.02) | 3.90 | 0.038 (**) |
| `antmaze` play | 0.24 (0.05) | 0.68 (0.05) | 6.22 | 0.003 (***) |
| `humanoidmaze` expert | 0.32 (0.03) | 0.56 (0.03) | 5.66 | 0.005 (***) |
| `humanoidmaze` play | 0.10 (0.02) | 0.49 (0.03) | 10.82 | 0.001 (***) |
| `cubesingle` | 0.74 (0.03) | 0.75 (0.02) | 0.28 | 0.797 (ns) |
| SAW vs SAW + TTT | | | | |
| `pointmaze` expert | 0.96 (0.01) | 0.99 (0.01) | 2.12 | 0.101 (ns) |
| `pointmaze` play | 0.84 (0.03) | 0.97 (0.01) | 4.11 | 0.038 (**) |
| `antmaze` expert | 0.97 (0.01) | 0.97 (0.01) | 0.00 | 1.000 (ns) |
| `antmaze` play | 0.98 (0.01) | 0.97 (0.01) | -0.71 | 0.519 (ns) |
| `humanoidmaze` expert | 0.85 (0.03) | 0.92 (0.03) | 1.65 | 0.174 (ns) |
| `humanoidmaze` play | 0.78 (0.02) | 0.84 (0.01) | 2.68 | 0.076 (*) |
| `cubesingle` | 0.73 (0.03) | 0.71 (0.03) | -0.47 | 0.662 (ns) |

*Table 8.* Welch's t-test for the main results reported in Table 1.

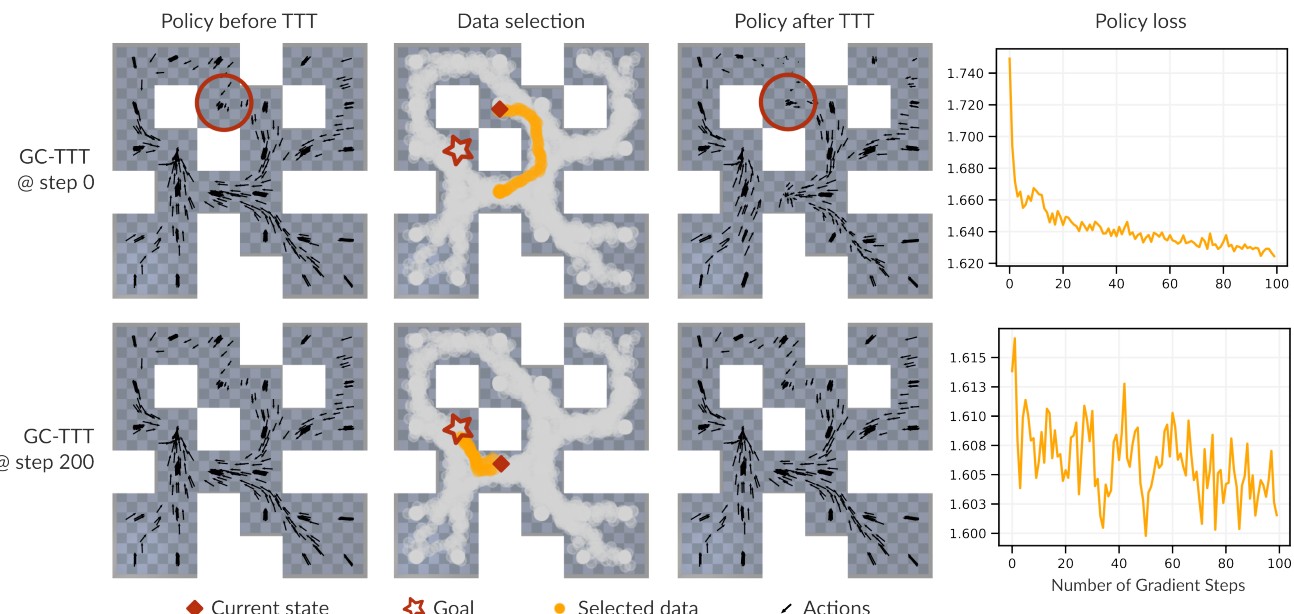

*Figure 9.* Visualization of GC-TTT in `pointmaze`. The first row represents the first iteration: the initial policy (left) is updated on selected data (center-left), resulting in the updated policy (center-right), which specializes on the current state-goal tuple (diamond and star). The actor loss over this adaptation process is reported on the right. The second row represents the second iteration of GC-TTT, which solves the task.

## E    Discussion with Respect to Graph-based Test-time Planning

TTGS (Opryshko et al., 2025) is a concurrent work that takes a different approach to test-time adaptation, namely explicit planning instead of in-weight updates. Empirically, TTGS can perform at least on par with GC-TTT on maze domains, as Opryshko et al. (2025) reports strong performance for larger mazes. However, TTGS is computationally less scalable: graph construction requires computing pairwise distances, and thus scales quadratically in the size of the dataset. The overhead of GC-TTT is instead due to gradient updates, while dependency on the size of the buffer is linear and almost negligible. Furthermore, while TTGS introduces several hyperparameters, which may be hard to tune (in particular, the threshold for graph pruning), GC-TTT is rather robust to hyperparameters, as reported in Appendix C.5. Overall, TTGS and GC-TTT are fundamentally orthogonal to each other: one solves a global problem through classical planning, while the other performs local improvements via gradient descent in parameter space. While both can be used for improving policy performance at test-time, we believe that the most exciting avenue of future work is to combine the two, and to allocate test-time compute adaptively to either component.

## F    Implementation Details

For environments and backbone algorithms, we adopt the default hyperparameters presented in OGBench (Park et al., 2025).

### F.1    Hyperparameters

GC-TTT introduces some additional hyperparameters. We keep the percentile fixed at $q = 0.05$ and tune the remaining ones, including the horizon $K$, the number of gradient steps $N$, and the fine-tuning learning rate $\alpha$ (see Table 9).

### F.2    Estimating FLOPs

Figure 5 (right) presents estimates of test-time compute (FLOPs) in its x-axis. In order to compute these estimates, we make the following simplifying assumptions:

- The input and output size of the policy is negligible with respect to its witdh $w$; hence, the number of sum/multiply operations for one forward pass is $C \approx 2nw^2 = 4w^2$, as the policy is an MLP with $n = 2$ hidden layers.

| | | pointmaze | | antmaze | | humanoidmaze | | cube | Fixed |
| | | expert | play | expert | play | expert | play | | |
|---|---|---|---|---|---|---|---|---|---|
| GC-BC + TTT (no critic) | $\alpha$ | $3e^{-4}$ | $3e^{-5}$ | $3e^{-4}$ | $3e^{-4}$ | $3e^{-4}$ | $3e^{-4}$ | $3e^{-5}$ | $3e^{-4}$ |
| | $N$ | 50 | 100 | 100 | 50 | 100 | 100 | 50 | 50 |
| | $K$ | 100 | 200 | 200 | 100 | 100 | 100 | 100 | 100 |
| GC-BC + TTT | $\alpha$ | $3e^{-5}$ | $3e^{-4}$ | $3e^{-4}$ | $3e^{-4}$ | $3e^{-4}$ | $3e^{-4}$ | $3e^{-4}$ | $3e^{-4}$ |
| | $N$ | 200 | 100 | 100 | 200 | 200 | 50 | 100 | 100 |
| | $K$ | 100 | 100 | 100 | 100 | 200 | 100 | 100 | 100 |
| GC-IQL + TTT (no critic) | $\alpha$ | $3e^{-4}$ | $3e^{-4}$ | $3e^{-4}$ | $3e^{-4}$ | $3e^{-5}$ | $3e^{-4}$ | $3e^{-5}$ | $3e^{-5}$ |
| | $N$ | 50 | 200 | 100 | 200 | 200 | 100 | 50 | 200 |
| | $K$ | 100 | 100 | 100 | 100 | 100 | 200 | 100 | 100 |
| GC-IQL + TTT | $\alpha$ | $3e^{-4}$ | $3e^{-4}$ | $3e^{-4}$ | $3e^{-5}$ | $3e^{-4}$ | $3e^{-5}$ | $3e^{-5}$ | $3e^{-4}$ |
| | $N$ | 100 | 200 | 100 | 200 | 50 | 100 | 50 | 100 |
| | $K$ | 200 | 100 | 200 | 100 | 200 | 100 | 100 | 100 |
| SAW + TTT (no critic) | $\alpha$ | $3e^{-5}$ | $3e^{-4}$ | $3e^{-5}$ | $3e^{-5}$ | $3e^{-5}$ | $3e^{-5}$ | $3e^{-4}$ | $3e^{-5}$ |
| | $N$ | 100 | 50 | 100 | 100 | 100 | 100 | 200 | 100 |
| | $K$ | 100 | 200 | 200 | 100 | 200 | 100 | 100 | 100 |
| SAW + TTT | $\alpha$ | $3e^{-4}$ | $3e^{-5}$ | $3e^{-5}$ | $3e^{-4}$ | $3e^{-4}$ | $3e^{-5}$ | $3e^{-5}$ | $3e^{-5}$ |
| | $N$ | 200 | 200 | 50 | 100 | 50 | 100 | 50 | 50 |
| | $K$ | 100 | 200 | 200 | 100 | 100 | 100 | 200 | 200 |

*Table 9.* Hyperparameters used in Table 1, with the last column containing fixed hyperparameters across environments used in Table 4.

- The cost of a forward pass does not depend on the batch size.

- A backward pass requires twice the compute as a forward pass.

Following from these assumptions, the cost for a single evaluation episode with 1000 steps is $C_{\text{no-TTT}} \approx 1000C = 4000w^2$. Considering the test-time training frequency $f$ and the number of gradient steps $m = 100$, the cost of the same operation with GC-TTT is $C_{\text{TTT}} = 1000f(1 + 6Cm) + 1000C$. The first term includes the cost of data selection (1 for the single forward pass required for computing values used in 8) and fine-tuning ($6Cm$, where we assume that the critic is the same size of the policy, and we need to compute gradients of the policy with respect to the critic's output). The cost of other operations not involving the neural network are not considered. Given the default width $w = 512$ we may then compute the compute cost without GC-TTT ($\approx 10^9$ FLOPs), and for test-time training frequencies $[1/1000, 1/500, 1/200]$ ($\approx 1.6 \cdot 10^9, 2.2 \cdot 10^9$ and $4 \cdot 10^9$ FLOPs, respectively). Given these increased compute budgets, we can finally solve for the values of $w$ necessary for meeting this compute cost without GC-TTT ($\approx 624, 732, 992$), which were used to obtain the grey curve in Figure 5 (right).

### F.3   Wall-clock Overheads

The operations required by GC-TTT have different latencies, which we measure on a RTX4090 GPU, and report in this Section. The relevance criterion requires $< 15s$ (once per episode, at the beginning); most of this delay is due to computation of $H$-step returns through a dataset-wide forward pass of the critic (skipped for the critic-free variant). Then, each time TTT is applied, evaluating the relevance criterion and filtering the dataset requires $< 1s$, while sampling each TTT batch and performing an optimization step require $< 10ms$ and $< 30ms$, respectively.

The total wall-clock overhead for GC-TTT at each each episode (considering $L = 1000$ environment steps, $N = 100$ gradient steps and a fine-tuning interval of $K = 100$) would be $15s + \frac{L}{K}(1s + N(10ms + 30ms)) \approx 65s$. Distributed over 1000 steps, this is an average cumulative overhead of $65ms$ per environment step, which is however not uniform: a significant part of this overhead is allocated before the first environment step, and the rest is concentrated during TTT.

