# OpenReview forum: "Test-time Offline Reinforcement Learning on Goal-related Experience"
_ICML.cc/2026/Conference — ICML 2026 regular_

### Official Review · Reviewer_cuvH · 2026-03-08

**Soundness:** 3
**Presentation:** 3
**Significance:** 2
**Originality:** 2
**Overall Recommendation:** 4
**Confidence:** 2

**Summary:**

This paper introduces GC-TTT, a test-time training framework for offline goal-conditioned RL. At evaluation time, the method selects sub-trajectories from the offline dataset that are both relevant to the current state and estimated to be good for the current goal, and then performs a few gradient steps to adapt the policy locally. The method is well evaluated on several OGBench navigation and manipulation tasks with multiple offline RL backbones.

**Compliance With Llm Reviewing Policy:**

Affirmed.

**Final Justification:**

the rebuttal addressed my concerns

**Key Questions For Authors:**

1. Why is there no direct comparison to TTGS or another frozen-policy test-time search/planning baseline, especially since this seems to be the most relevant competing direction?
2. In the experiments, how much does the method rely on environment-provided distance versus a learned distance / quasimetric? What happens if the environment distance is removed entirely? What would “local relevance” actually mean in reasoning or more complex settings?
3. What is the intended advantage of policy fine-tuning over frozen-policy graph search at test time: better final performance, better asymptotic scalability, or something else?

**Limitations:**

See weakness

**Strengths And Weaknesses:**

## Strengths

1. The paper studies a meaningful problem. In offline goal-conditioned RL, training is amortized over many goals, while evaluation only cares about one goal at a time, so test-time specialization is a natural direction.
2. The method is conceptually clear. The relevance + optimality data selection rule is easy to understand, and the receding-horizon adaptation scheme is also well motivated.

## Weaknesses

1. **The novelty mostly comes from bringing TTT into offline GCRL.**
   The core idea of the paper feels like a reasonable transplantation of TTT into GCRL. The closest related work already uses extra test-time adaptation in offline GCRL, but by keeping the policy frozen and performing planning / search instead. The paper does not fully clarify why parameter adaptation should be viewed as a substantially different or stronger direction.

2. **The comparison to the closest related methods, especially TTGS, is not sufficient.**
   The paper itself discusses test-time graph search / subgoal search as a very close alternative. With test-time computation, it is important to know whether it is better spent on policy fine-tuning or on frozen-policy search/planning. Without a head-to-head comparison, I am not convinced that the extra cost and complexity of test-time fine-tuning are justified, especially given possible concerns about adaptation speed, optimization instability, and generalization.

3. **The method relies heavily on a local relevance assumption.**
   A key component of GC-TTT is selecting data that are close to the current state, using either an environment-provided distance or a learned quasimetric. This is very natural in maze / navigation style tasks. But in more general tasks this assumption is much less obvious, especially in:
   - high-dimensional visual observation settings,
   - more complex manipulation problems where local similarity may not reflect reachability,
   - domains such as reasoning or NLP.
   As a result, the current evidence mainly supports the method in OGBench-style goal-reaching tasks, not yet in broader decision-making settings.

---

> ### Author Rebuttal · Authors · 2026-03-30
>
> Thank you for your detailed and thoughtful comments. We understand the points you raised, and we will address each one individually.
>
> > W2/Q1: Why is there no direct comparison to TTGS or another frozen-policy test-time search/planning baseline?
>
> Thanks for bringing this point forward; we are happy to further clarify this in our writing. To the best of our knowledge, TTGS [1] is a concurrent work that takes a radically different approach to adaptation, namely explicit planning instead of in-weight updates. This comes with advantages and disadvantages. Empirically, we expect TTGS to perform at least on par with GC-TTT on maze domains, as [1] reports strong performance for larger mazes. However, TTGS comes with limitations, as described in the next answer.
>
> > W1/Q3: What is the intended advantage of policy fine-tuning over frozen-policy graph search at test time?
>
> While TTGS is a strong method, it has three limitations in our opinion. The main limitation is computational: graph construction requires computing pairwise distances, and thus scales quadratically in the size of the dataset. The second limitation is practical: TTGS searches for a global solution (i.e. an explicit sequence of subgoals), and it thus requires a dataset with balanced coverage over the entire state space. A third, lesser limitation is that TTGS introduces several hyperparameters, which may be hard to tune (in particular, the threshold for graph pruning).
>
> We believe that GC-TTT represents an alternative and orthogonal framework for test-time adaptation, which intends to approach the performance of specialized graph search algorithms while remaining simpler and more scalable. In particular, the main advantage is computational: data selection procedure incurs a linear cost with respect to the dataset size. Second, as the data selection procedure acts locally, globally uniform coverage is not necessary. Third, GC-TTT is not particularly sensitive to hyperparameters (see response to rDQJ, W1 for empirical evidence). Furthermore, due to its similarity with test-time scaling methods in broader deep learning settings, we expect that GC-TTT will enable external advancements to be applied to RL problems.
>
> Overall, TTGS and GC-TTT are fundamentally orthogonal to each other: one solves a global problem through classical planning, while the other performs local improvements via gradient descent in parameter space. While both can be used for improving policy performance at test-time, we believe that the most exciting avenue of future work is to combine the two, and to allocate test-time compute adaptively to either component.
>
> We understand that this discussion helps in contextualizing our method, and we will integrate this discussion in the next revision.
>
> > W3/Q2: how much does the method rely on environment-provided distance versus a learned distance / quasimetric? What happens if the environment distance is removed entirely? What would “local relevance” actually mean in reasoning or more complex settings?
>
> Thank you for raising this great point. The “environment distance” is not a strict requirement for GC-TTT. As we show in Appendix C.3, local relevance may be learned end-to-end, and is in particular well-captured by a thresholded goal-conditioned value function. Intuitively, in goal-conditioned settings, value functions can be mapped to temporal distances over the MDP’s dynamics [3], and states $s$ for which $V(s, g)$ is above a given threshold can be interpreted as “local” to $g$. As a result, we can formulate a more general variant of GC-TTT that may be applied when an “environment distance” is not available. Empirical validation (Table 3) confirms that performance is maintained, even when using learned (and thus noisy) value estimates.
>
> In more complex settings, the relevance of state $s’$ with respect to state $s$ can still be defined as a low number of expected steps required by a (possibly optimal) policy to transition from $s$ to $s’$ (i.e, a goal-conditioned value function). For instance, in the suggested context of mathematical reasoning in natural language, we can say that a reasoning trace deriving Lemma $g$ from Lemma $s’$ is relevant for the current Lemma $s$ if $s’$ can be derived from $s$ in a few tokens. While learning value functions over text tokens remains challenging, relevance can in principle be captured by a goal-conditioned value function $V(s,s’)$.
>
> ---
> We hope we were able to answer each constructive comment in a precise and clear way. We would like to ask the reviewer whether their concerns have been addressed in full or partially, and we remain available for further discussion in any case.
>
> ---
> [1] Opryshko et al., Test-Time Graph Search for Goal-Conditioned Reinforcement Learning, arXiv 2026
>
> [2] Bagatella et al., Goal-conditioned Offline Planning from Curious Exploration, NeurIPS 2023
>
> [3] Nasiriany et al., Planning with Goal-Conditioned Policies, NeurIPS 2019

---

> > ### Author Rebuttal · Reviewer_cuvH · 2026-04-03
> >
> > The rebuttal partially addresses my concerns. It clarifies the intended positioning of GC-TTT relative to frozen-policy search methods and gives a more convincing response on replacing environment distance with learned local relevance. However, the lack of a direct comparison to TTGS or another strong test-time planning baseline remains an important gap, and the discussion of broader applicability beyond OGBench-style goal-reaching tasks is still mostly conceptual.

---

> > > ### Author Response · Authors · 2026-04-07
> > >
> > > Thank you for the acknowledgement of our rebuttal. While TTGS constitutes concurrent work, its codebase is publicly available, and we were thus able to provide a direct comparison, as requested. We are happy to report the following in the next revision.
> > >
> > > Our evaluation adopts GC-IQL as the underlying RL algorithm, as both TTGS and GC-TTT consider it. We evaluated TTGS over the environments from Table 1; for each environment, we swept over the full range of hyperparameters proposed for TTGS in Table 4: $\tau=[6, 12, 24, 36]$ and $T=[12, 24, 48, 72]$. We report performance for the best set of hyperparameters in each environment, and discuss how these results empirically confirm our previous comments.
> > >
> > > | | pointmaze expert | pointmaze play | antmaze expert | antmaze play | humanoidmaze expert | humanoidmaze play | cube single |
> > > |-|-|-|-|-|-|-|-|
> > > |TTGS|$\mathbf{88\pm0.07}$|$\mathbf{61\pm0.17}$|$\mathbf{89\pm0.02}$|$\mathbf{77\pm0.12}$|$\mathbf{59\pm0.10}$|$22\pm10$|$44\pm0.10$|
> > > |GC-TTT|$60\pm0.00$|$41\pm0.03$|$\mathbf{87\pm0.03}$|$\mathbf{73\pm0.06}$|$\mathbf{57\pm0.01}$|$\mathbf{53\pm0.04}$|$\mathbf{75\pm0.04}$|
> > >
> > > > we expect TTGS to perform at least on par with GC-TTT on maze domains
> > >
> > > We find that this is the case, at least in *pointmaze* and *antmaze*, confirming that, while adopting a radically different approach, TTGS also constitutes a strong method for scaling at inference.
> > >
> > > > TTGS introduces several hyperparameters, which may be hard to tune
> > >
> > > This can be observed in the performance of TTGS on *humanoidmaze*, which is surprisingly lower than what reported for mazes of larger sizes. We believe that this indicates that substantially different hyperparameters may be needed for this maze size. We attempted a slightly enlarged sweep ($\tau \in [3, 48]$, $T \in [6, 96]$), which however yielded no significant improvement. TTGS also fails in manipulation in the current settings.
> > >
> > > > The main limitation is computational: [TTGS] requires computing pairwise distances, and thus scales quadratically in the size of the dataset
> > >
> > > We remark that local policy fine-tuning and graph search-based algorithms have fundamentally different computational requirements. We can observe this by measuring the single-episode wall-clock overhead of TTGS and GC-TTT as the size of the dataset sample ($N$) grows:
> > >
> > > ||$N=4K$|$N=10K$|$N=100K$|$N=1M$|
> > > |-|-|-|-|-|
> > > |TTGS| $32$s | $124$s | OOM | OOM |
> > > |GC-TTT| $50$s | $51$s | $52$s | $65$s |
> > >
> > > The overhead of TTGS is dominated by graph construction, whose complexity is quadratic in the size of the dataset sample, resulting in OOM errors and time-outs for moderately large sample sizes. The overhead of GC-TTT is instead due to gradient updates, while dependency on the size of the buffer is linear and almost negligible.
> > > - - -
> > > We remark that gradient-based and planning-based methods can be considered to be orthogonal, and may be freely combined. We nevertheless agree that it is important to provide a direct comparison between approaches, highlighting both upsides and downsides. We hope that this discussion may lift the remaining concerns, and we are happy to include it in the revision.

---

### Official Review · Reviewer_q9T8 · 2026-03-13

**Soundness:** 2
**Presentation:** 2
**Significance:** 2
**Originality:** 2
**Overall Recommendation:** 4
**Confidence:** 4

**Summary:**

The paper proposes goal-conditioned test-time training (GC-TTT) for improving offline reinforcement learning, by sampling goal-related experience from offline dataset and using it to fine-tune the policy at test time. In order to sample these experiences, the approach picks them using proximity to the current state, and optimality by an H-step return estimate using a learned critic. They also propose a critic-free variant using expert data. The experiments show that building on approaches like GC-BC, GC-IQL, and SAW, their approach is able to outperform the baselines, while adding test-time compute. The updates are done every K steps at inference time.

**Compliance With Llm Reviewing Policy:**

Affirmed.

**Final Justification:**

The authors presented results with additional seeds and added relevant clarification on using a learned value function as an alternative for the local distance estimate. I will keep a positive score.

**Key Questions For Authors:**

1. On high dimensional tasks, what methods can be used as alternatives for selecting goal-related data from offline pre-training data?
2. Can the authors comment on how the value of K impacts the performance?
3. In optimality based data selection, how robust is the method to incorrect value function?

**Limitations:**

Please see weaknesses above.

**Strengths And Weaknesses:**

Strengths:
1. The paper demonstrates substantial gains over prior backbones empirically.
2. The methodology for extracting goal-related experiences is simple and easy to imeplement.
3. The method can be combined with various offline RL algorithms (GC-BC, GC-IQL, SAW).

Weaknesses:
1. The approach has limited novelty, as it is basically a data-selection procedure for training the policy using goal-relevant data.
2. The motivation behind sampling goal related experience experience is weak and only empirically justified.
3. The experiments are done over only 3 seeds, which could lead to noisy results. The authors should consider extending the experiments to at least 5 seeds.
4. Calculating proximity to select goal-related experience may become hard on high-dimensional tasks like vision based tasks.
5. The updates are done after every K timesteps, which is computationally expensive.

---

> ### Author Rebuttal · Authors · 2026-03-30
>
> We would like to thank the reviewer for the constructive and actionable comments; we believe we can address them directly.
>
> > W1: The approach has limited novelty
>
> We respectfully disagree. While data selection is at the core of our framework, a key contribution is when and how this data selection is applied, as described in Section 4.2. While recent works have suggested local gradient updates for policy improvement [1], they still require up to 2M gradient steps; our work is, to the best of our knowledge, the first to directly focus on efficient gradient-based test-time training in GCRL. Moreover, we provide a systematic study of performance across environments and backbones, as well as an ablation of each moving part.
>
> > W2: The motivation behind sampling goal related experience experience is weak and only empirically justified
>
> We believe that our work remains grounded from a formal perspective. In particular, as the radius $\epsilon$ of the relevance criterion (Eq. 6) shrinks, the combination of data selection and gradient updates to the policy may be seen as a generalized policy improvement operator [2] acting on the policies generating the data. As such, if carried out exactly, GC-TTT could be shown to match or exceed the performance of the best policy in the dataset. We are happy to further formalize this derivation in our writeup given more space.
>
> > W3: only 3 seeds
>
> Thank you for this actionable suggestion. Through the course of this rebuttal we were able to run an additional 2 seeds for each experiment in Table 1, for a total of 5, confirming our results (see response to LV1W, W1).
>
> > W4, Q1: calculating proximity to select goal-related experience may become hard on high-dimensional tasks
>
> The local distance used in Eq. 6 may not always be available, especially in high-dimensional spaces in which e.g. the L2 distance may not be meaningful. This limitation can, however, be circumvented, as we explore in Appendix C.3. We demonstrate that the local distance estimate can be replaced with a thresholded learned value function without inducing significant drops in performance. As a result, our framework may be further scaled, since value functions have been shown to be learnable on high-dimensional tasks [3].
>
> > W5, Q2: Can the authors comment on how the value of K impacts the performance?
>
> We ablate the role of $K$ in Figure 5 (middle), which reports the performance as $K$ decreases - and the TTT frequency thus increases. The performance of GC-TTT already exceeds that of the backbone algorithm for the highest values of $K$, for which the agent is only updated once, at the start of the episode (leftmost points in the plot). As more compute is available and $K$ grows smaller, success rates increase steadily before plateauing. We finally note that performance tends to improve further with more compute in more complex environments (e.g., *antmaze* vs *pointmaze*).
>
> > Q3: how robust is the method to incorrect value function?
>
> Thank you for this interesting question. While GC-TTT relies on an estimated value function for data selection, our experiments suggest that it is rather robust to inaccuracies. In particular, we find that ranking subtrajectories does not require the same degree of accuracy that policy extraction does. We can observe this, for instance, in Fig. 9. The value function learned by GC-IQL, and later used by GC-TTT is not accurate enough to extract a successful policy in this *pointmaze* task (first column). However, it correctly identifies promising subtrajectories (second column).
> To provide further evidence, we have additionally run an ablation measuring the performance of GC-TTT in *antmaze*-play as noise is injected into the value estimates for optimality-based data selection (Eq. 7, 8). Upon computing value estimates (Eq. 7), we compute the average absolute value $\bar V$ across a batch, and corrupt each value estimate with Gaussian noise with standard deviation $\sigma=\alpha \bar V$. When increasing $\alpha$, we observe that performance does not degrade significantly for modest perturbations, and only does so under strong perturbations ($\alpha=1$). We will include this in the revision.
> |$\alpha=0$|$\alpha=0.1$|$\alpha=1$|
> |-|-|-|
> |$0.73\pm0.06$|$0.78\pm0.03$|$0.31\pm0.02$|
>
> ---
> Thank you for taking the time to review our submission and for providing actionable feedback. We hope we were able to address each point, in particular by highlighting the novelty and grounding of our method (W1-2) and by answering remaining weaknesses and each question with empirical evidence. In case any concern still stands, we are happy to continue this constructive conversation further.
>
> ---
> [1] Park et al., Is Value Learning Really the Main Bottleneck in Offline RL?, NeurIPS 2024
>
> [2] Barreto et al., Successor features for transfer in reinforcement learning, NeurIPS 2017
>
> [3] Mnih et al., Human-level control through deep reinforcement learning, Nature 2015

---

> > ### Author Rebuttal · Reviewer_q9T8 · 2026-04-04
> >
> > I thank the authors for their detailed responses. I appreciate the additional seeds experiments and clarification on using a learned value function as an alternative for the local distance estimate. In line with reviewer cuvH's observation, I encourage the authors to include comparisons to test-time baselines in the final version. Based on this, I am increasing my score.

---

### Official Review · Reviewer_rDQJ · 2026-03-13

**Soundness:** 3
**Presentation:** 3
**Significance:** 2
**Originality:** 3
**Overall Recommendation:** 5
**Confidence:** 4

**Summary:**

This paper addresses offline goal-conditioned RL at evaluation time and introduces Goal-Conditioned Test-Time Training (GC-TTT), which periodically updates the actor using offline sub-trajectories selected for local relevance and goal-conditioned value. The method combines relevance/optimality filtering from bootstrapped returns, and horizon adaptation with resets to the original policy weights.
Empirically, it consistent gains on four OGBench tasks across three backbones
(GC-BC, GC-IQL, SAW).

**Compliance With Llm Reviewing Policy:**

Affirmed.

**Final Justification:**

The rebuttal directly addressed my concerns and, I believe, improved the submission quality. Consequently I decided to improve my score.

**Key Questions For Authors:**

**Questions and Recommendations:**
- How much of the reported improvement remains if the method uses one global set of TTT hyperparameters, fixed once and not retuned for each environment or backbone ?
- Could the authors provide a stronger compute-matched analysis beyond antmaze-play, maybe including wall-clock comparisons rather than only approximate FLOP estimates ?
- How often does GC-TTT degrade individual episodes or create instability, even when average performance improves ?
- What is the empirical relationship between |D_rel(s)| / local dataset density and performance gain ? This seems central to understanding where the method can succeed or fail.
- Would a simpler baseline that only reweights or replans over the selected data, without gradient based actor updates, recover much of the gain ?

**Minor Comments:**
- There appears to be a typo in Related Work with MPDs that should be MDPs.
- There is a duplicated phrase in the Related Work “improve performance the performance”.

**Limitations:**

yes

**Strengths And Weaknesses:**

**Strengths:**
- The method is well-structured and easy to integrate. It builds on a standard actor-training objective and adds only a test-time trajectory-selection stage plus periodic local refinement, making the approach straightforward to attach to existing offline GCRL backbones.
- Main experimental evaluation is reasonable for a method paper.
- The paper does a good job isolating the key mechanism. Figure 5 shows that random selection, relevance-only, and optimality-only all fall short of the full relevance + optimality rule.
- The critic-free version is a practical extension for expert-data settings, and the appendix is useful in that it includes fixed hyperparameters results and additional-backbone analyses rather than only presenting the most favorable tuned configuration.



**Weaknesses:**
- Main results rely on environment-specific tuning of test-time learning rate, gradient steps, and horizon, which weakens the plug-and-play interpretation. Fixed hyperparameter results are provided, but only in the appendix.
- The “minimal compute cost” framing feels too strong. Section 4.3 and Appendix E.3 report ~ 70s per 1000-step episode for the critic-based setting, with uneven latency concentrated around adaptation phases.
- The matched compute argument is limited as it covers only antmaze-play and depends on simplified FLOP estimates.
- The empirical scope is still modest relative to the broader framing, as the study is restricted to OGBench medium tasks, four fixed goals per environment, and 3 seeds, with no direct evaluation under weaker local coverage, larger goal spaces, or more severe distribution shift.
- Broader applications of the method depends on having either a useful local distance or sufficiently accurate value estimates for trajectory selection, and the paper only partially probes these dependencies.

---

> ### Author Rebuttal · Authors · 2026-03-30
>
> Thank you for the thorough review and thoughtful comments. We believe we can improve our submission by answering each question, and directly applying your suggestions.
>
> > W1, Q1: one global set of TTT hyperparameters
>
> As you point out, we included results without environment-specific tuning in the Appendix to demonstrate robustness to hyperparameters. We can strengthen this claim by fixing hyperparameters across backbones as well, which we have now done. For $lr=3e^{-4}$, $N=100$, $K=100$, we observe that average performance of GC-TTT across environments drops minimally: $0.58 \to 0.56$, $0.63 \to 0.59$, $0.92 \to 0.89$ with GCBC, GCIQL and SAW as backbones, respectively. Due to space limitations, we can report full results in the next answer or revision.
>
> > W2: The “minimal compute cost” framing feels too strong
>
> We refer to the compute cost as “minimal” as the average inference frequency remains near real-time. If better fitting, we can weaken this claim, referring to “modest compute costs” throughout the writeup.
>
> > W3, Q2: The matched compute argument is limited [...]
>
> Thank you for raising this important point. Following this suggestion, we have repeated the analysis across *pointmaze* and *humanoidmaze*. We report similar trends to those in antmaze (Figure 5, right):
>
> ||no TTT (1x)|K=1000 (1.6x)|K=500 (2.2x)|K=200 (4x)|
> |-|-|-|-|-|
> |pointmaze|0.24±0.07|0.25±0.06|0.38±0.06|0.40±0.04|
> |humanoidmaze|0.08±0.03|0.20±0.01|0.29±0.01|0.47±0.05|
>
> ||w=512 (1x)|w=624 (1.6x)|w=732 (2.2x)|w=992 (4x)|
> |-|-|-|-|-|
> |pointmaze|0.24±0.07|0.16±0.02|0.17±0.06|0.25±0.04|
> |humanoidmaze|0.08±0.03|0.11±0.01|0.15±0.02|0.22±0.02|
>
> As for units, we have chosen FLOPs as they abstract away hardware-specific quirks. If we were to use wall-clock time on our hardware, the curve corresponding to GC-TTT would slightly stretch on the $x$-axis; however, as the model-scaling shows weak-to-no improvements at modest model sizes, we do not expect conclusions to change.
>
> > W4: The empirical scope is still modest
>
> The main goal of our evaluation is that of demonstrating performance on an established benchmark, as it was designed. We were, however, able to extend our main results to include 2 extra seeds (for a total of 5). We confirm existing results (see response to LV1W, W1).
>
> > W5: the method depends on having either a useful local distance or sufficiently accurate value estimates
>
> We generally agree; however, we believe that GC-TTT demonstrates robustness in this regard. In Appendix C.3 we demonstrate that a local distance may be replaced by a thresholded value estimate. During this rebuttal, we have additionally studied the injection of noise in value estimates (antmaze-play, GC-IQL). Upon computing value estimates (Eq. 7), we compute the average absolute value $\bar V$ across a batch, and corrupt each estimate with Gaussian noise with standard deviation $\sigma=\alpha \bar V$. For $\alpha=0$, GC-TTT achieves a success rate of $0.73\pm0.06$. When increasing $\alpha$, we observe that performance does not degrade significantly for modest perturbations, and only does so under strong perturbations ($\alpha=1$). We will include this in the revision.
> |$\alpha=0$|$\alpha=0.1$|$\alpha=1$|
> |-|-|-|
> |0.73±0.06|0.78±0.03|0.31±0.02|
>
> > Q3: how often does GC-TTT degrade individual episodes?
>
> We find that performance degradation is rare in GC-TTT for two reasons. First, the agent is updated for a modest number of steps, and the policy does not change dramatically. Second, GC-TTT is naturally conservative: when the agent’s state is out of distribution, and updates would be noisy, the relevance criterion (Eq. 6) largely returns empty datasets, and the base policy is left untouched, preventing compounding instabilities.
>
> > Q4: [...] empirical relationship between |D_rel(s)| / local dataset density and performance gain
>
> As discussed above, when the local dataset density is low, the number of relevant samples decreases, and the updates to the policy are either noisy or rejected, resulting in more modest performance gains. Empirically, we find that in a representative setting (*antmaze* play, GC-IQL), the average size of $D_{rel}$ is ≈$48$% larger for episodes that result in success as opposed to failures, which empirically confirms our argument.
>
> > Q5: Would a simpler baseline that only reweights or replans over the selected data [...] recover much of the gain ?
>
> We have evaluated a baseline that averages actions over the selected dataset, and replays them in an open loop fashion. We find that this method underperforms the base policy and achieves a near-zero success rate in *antmaze*, as it loses reactivity and may not consider state dependencies.
>
> > C1, C2: [typos]
>
> Thank you for catching these typos, we are happy to fix them.
>
> ---
> We would like to thank the reviewer for the positive and constructive feedback. We hope we were able to address your main concerns and improve our submission given the limited space. We remain available for further discussion.

---

> > ### Author Rebuttal · Reviewer_rDQJ · 2026-04-04
> >
> > Thanks for the clarification, which solves my concerns. I'm looking forward to seeing the updated version of the manuscript. I will raise my score.

---

### Official Review · Reviewer_LV1W · 2026-03-15

**Soundness:** 3
**Presentation:** 4
**Significance:** 3
**Originality:** 3
**Overall Recommendation:** 5
**Confidence:** 4

**Summary:**

This paper proposes goal-conditioned test-time training in an offline reinforcement learning setting. The approach begins by pertaining a model on an offline dataset containing trajectories that achieve a diverse set of goals. At tea time, the model is dynamically fine-tuned using data from the pertaining dataset that is most relevant to the current goal. Relevance is determined by extracting sub-trajectories and evaluating their H-step return with respect to the target goal. These candidates are then further filtered using a q-th percentile threshold to retain only the most optimal sub-trajectories for adaptation.

**Compliance With Llm Reviewing Policy:**

Affirmed.

**Final Justification:**

Concerns are addressed and clearly answered.

**Key Questions For Authors:**

* How is "significant" performance gain quantified? Reporting statistical significance test would help.
* In some environments, GC-BC or GC-IQL-DDPG fails to reach the performance level of SAW. What factors contribute to this difference?
* Do the authors plan to release the code to support reproducibility?
* In Figure 9, are the results of N gradient steps and K training frequency combined?
* How does the method behave under OOD test scenarios? If no relevant samples are found, does GC-TTT effectively avoid mis-adaptation?

**Limitations:**

Yes

**Strengths And Weaknesses:**

### Strengths

* The paper Is well written and clearly presented.
* Good selection of baselines and experiments clearly display the benefit of TTT.
* The proposed work is significant to address test time generalization problems in offline reinforcement learning.

### Weakness

* Only 3 seeds for experiments seems low.
* It is unclear how the proposed TTT performs as data quality degrades, since the experiments are conducted only on medium-quality datasets.

---

> ### Author Rebuttal · Authors · 2026-03-30
>
> Thank you for your positive assessment and your thoughtful comments. We are happy to address each of them.
>
> > W1: [...] only 3 seeds for experiments seems low
>
> Following your suggestion, we were able to run 2 additional seeds for the main results (Table 1), for a total of 5. The updated results in Table 1 are consistent with the previous ones, with mean estimates of performance shifting only slightly, as expected. Concisely, the average success rate with vs without TTT across environments remains largely unchanged: $0.23$ vs $0.58 \to 0.22$ vs $0.56$ for  GCBC, $0.39$ vs $0.63 \to 0.39$ vs $0.61$ for GCIQL, $0.86$ vs $0.92  \to 0.87$ vs $0.91$ for SAW. Given space limits, we can report the updated Table 1 in the next response, or in the next revision directly.
>
> > W2: [...] the experiments are conducted only on medium-quality datasets
>
> In adopting the standard evaluation protocol in OGBench, we have focused our evaluation across two types of datasets: *navigate* and *stitch*. While both contain relatively high-quality trajectories, they are fundamentally different: *navigate* datasets contain demonstrations for evaluation tasks, while *stitch* datasets explicitly avoid those demonstrations, and instead contain trajectories that are not directly related to evaluation tasks. We thus believe that the consistent performance of GC-TTT in these two regimes suggests a degree of robustness to different data sources.
>
> > Q1: How is "significant" performance gain quantified? Reporting statistical significance test would help
>
> We have originally marked improvement as significant when the confidence intervals denoted by standard errors do not overlap (e.g. in Table 1). Following your suggestion, we have additionally run a Welch's t-test in the course of this rebuttal, confirming that performance improvements are significant ($p<0.1$) in $14/21$ environment/backbone combinations (most non-significant improvements occur with SAW as a backbone, which already masters some of the tasks). We will add the full Table to the next revision.
>
> > Q2: GC-BC or GC-IQL-DDPG fails to reach the performance level of SAW. What factors contribute to this difference?
>
> This trend is generally consistent with results reported in SAW [1]: the implicit hierarchy enforced by SAW during training results in an effectively shortened decision horizon, and eventually in better policies. Despite the lower performance of GC-BC or GC-IQL, these methods are simpler and well-established; we include them in our evaluation to confirm that GC-TTT remains agnostic to the underlying algorithm.
>
> > Q3: Do the authors plan to release the code [...] ?
>
> In order to support reproducibility, we have released our code on the anonymous website, which is linked on line 284. We are happy to highlight this by a mention in the introduction.
>
> > Q4: In Figure 9, are the results of N gradient steps and K training frequency combined?
>
> Fig. 9 displays both in separate ways. Each row represents a single iteration of TTT: the policy depicted in the third column is the result of $N=100$ gradient steps on the policy depicted in the first column. A larger number of gradient steps would induce greater differences between policies.
>
> At the same time, the first row represents the first TTT iteration (right before the agent takes any step in the environment), while the second row represents the second TTT iteration, which occurs after $1/K=200$ steps. In particular, the second column displays the agent’s current state and selected data. We can see that the current states are different between the first and the second row, as 200 environment steps occurred in between them. A higher TTT frequency $K$ would be visible exactly here, as it would likely result in the current states being closer to each other. We are open to any suggestions on additional visualizations we may provide.
>
> > Q5: If no relevant samples are found, does GC-TTT effectively avoid mis-adaptation?
>
> Thank you for the question, which we believe highlights a natural robustness of the data selection property in GC-TTT. If no relevant samples are found, the TTT dataset $\mathcal{D}_\text{rel}$ is empty (Eq. 6), and no gradient update is applied. Thus, as long as local relevance is computed accurately, instead of training on potentially misleading data, GC-TTT naturally rejects updates and avoids mis-adaptation.
>
> ---
> We sincerely appreciate your review, and hope that each of your comments was addressed. We are happy to further discuss any potential follow-up.
>
> ---
> [1] Zhou et al., Flattening Hierarchies with Policy Bootstrapping, NeurIPS 2025

---

> > ### Author Rebuttal · Reviewer_LV1W · 2026-04-03
> >
> > Thank you for clear answers. I will keep my positive scores.

---

### Decision · Program_Chairs · 2026-04-30

**Decision:**

Accept (regular)

**Comment:**

This paper presents GC-TTT, a test-time training framework for offline goal-conditioned RL that adapts policies using goal-related experience selected from the offline dataset. The reviewers found the problem meaningful, and the method technically sound. The rebuttal further addressed concerns regarding robustness, computational cost, and comparison to related test-time approaches. Overall, I find the paper solid and relevant, and I recommend acceptance.